# Assessing the feasibility and impact of specially adapted exercise interventions, aimed at improving the multi-dimensional health and functional capacity of frail geriatric hospital inpatients: protocol for a feasibility study

Paul Doody [iD],[1] Janet M Lord,[2,3] Carolyn A Greig,[1,2,3] Anna C Whittaker[1,4]

For numbered affiliations see end of article.

**Correspondence to**
Paul Doody;
p.d.doody@bham.ac.uk

## ABSTRACT

**Background** Frailty is a common and clinically significant condition in older adults, predominantly due to its association with adverse health outcomes such as hospitalisation, disability and mortality. Exercise interventions have been shown to be a beneficial treatment for frailty. However, more high-quality studies are needed to assess the feasibility and impact of these interventions in frail geriatric populations within different settings, and their impact on broader aspects of health and well-being.

**Methods and analysis** This study will use a 2-week, interventional, independent measures research design in order to assess the feasibility and impact of two specially adapted exercise training interventions (a specially adapted resistance training intervention, and Move It Or Lose It: an established community-based exercise intervention for older adults) aimed at improving the multidimensional health and functional capacity of frail geriatric hospital inpatients.

**Ethics and dissemination** This study has received a favourable ethical opinion by the Coventry and Warwickshire NHS Research Ethics Committee and sponsorship by the University of Birmingham after review by the sponsors research governance office. The findings will be disseminated through publication in open access scientific journals, public engagement events, online via social media, conference presentations and directly to study participants on request.

**Trial registration number** NCT03141866

## Strengths and limitations of this study

► Mixed-methods feasibility study employing both quantitative and qualitative research methodologies.
► Specially adapted exercise interventions for frail geriatric populations.
► Difficult to reach (and often excluded) participant population.
► Single-site study.

## BACKGROUND

Frailty is a common and clinically significant condition within geriatric populations,[1] predominantly due to its association with adverse health outcomes such as hospitalisation, disability and mortality.[1–6] Although the exact prevalence of frailty within this population is poorly defined due to the lack of a single standardised operational definition for the classification of frailty, it is generally believed that the prevalence of frailty among community-dwelling older adults ranges between 7.0% and 16.3%.[1 7] Presently, there are no well-evidenced, pooled estimates of the overall prevalence of frailty within geriatric hospital inpatients. However, there are several studies which have primarily aimed to produce estimates of the prevalence of frailty within this population.[8–15] Although there is an evident need for more robust research to thoroughly assess the overall prevalence of frailty within this setting, through analysis of these existing studies, dependent on the criteria utilised, the prevalence of frailty in geriatric hospital inpatients ranges from ~27% to 94%. In the five of the eight studies which utilised the Fried Frailty phenotype[2] as the operational definition of frailty, there is a narrower range (27%–48.5%), with a mean prevalence of frailty across the five studies of 37.5%±6.76%.[8 9 11 12 16]

Although there is no one standardised and universally utilised operational definition of frailty, one of the most commonly utilised is the Fried frailty phenotype.[2] This proposes that frailty be defined as a clinical syndrome in which three or more of the five following criteria are present: unintentional weight loss (≥10lbs in the last year), self-reported exhaustion, weakness (grip strength), slow walking speed and low levels of physical activity.

Exercise interventions have been proposed as potentially offering the best form of treatment for frail older adults[17]; with exercise shown to be a significantly beneficial treatment for this population with regard to multiple components of health, and even shown to mediate the reversal of frailty in some cases.[18–21] However, while there is evidence of the benefits of exercise relating to the prevention, treatment and reversal of frailty, it is universally noted that there needs to be more high quality studies within this area to truly assess the impact of exercise in frail geriatric populations within different settings, particularly relating to its effects on broader aspects of health and well-being.[1]

This present study will assess the feasibility and efficacy of short duration (2 weeks), intensive (5 days/week), specially adapted exercise interventions within a delayed transfer of care hospital ward setting. Feasibility will relate to the eight main areas of focus for feasibility studies,[22] while efficacy will be assessed through limited efficacy testing of the impact of the interventions on the secondary dependent variables relating to multidimensional health and functional capacity.

Such research is very timely and pertinent, as current demographic trends indicate that by the year 2030 almost one in six of the European population will be aged 60 years or over, and the number of older people will grow to 247 million by 2050, representing a 35% increase from 2017, with one in four older adults being over 85 years by 2040.[23] This coupled with continual progressive declines in the rate of physical activity, at all stages of the lifespan,[24] leaves the population particularly susceptible to the development of disease and comorbidities associated with a lack of physical activity and an increase in sedentary behaviour.[25] Moreover, acute hospital admission for older adults are associated with further loss of physical activity and represent a period of increased susceptibility to sarcopenia and frailty.[26] Frailty is associated with longer stay and increased rates of mortality in hospitalised older adults, as well as serving as a predictor of readmission.[12 27] Therefore, there is an urgent need to examine the effect of such interventions within this setting, and whether these interventions can be employed to improve various aspects of health in frail older populations in inpatient hospital ward settings, as well as their efficacy in specifically treating, preventing and reversing frailty. Preliminary research has shown some success in the implementation of exercise interventions to reverse functional decline in general geriatric inpatient populations[28]; however, to the authors' knowledge, this present study is the first to attempt such an intervention in frail delayed transfer of care patients.

## METHODS AND ANALYSIS
### Aims and objectives
The primary aim of this study is to assess the feasibility of a proposed future trial in this setting, which aims to assess the impact of specially adapted exercise interventions on the physiological, psychological, cognitive, social and emotional health, and functional capacity of frail geriatric populations within a hospital ward setting; recognising health as a multifactorial concept incorporating multiple inter-related dimensions. The secondary aim of this feasibility study is to assess the potential efficacy of the interventions on the primary dependent variables of the proposed future clinical trial within this setting.

The primary and secondary aims of this study will be achieved through the sequential achievement of the following objectives: (1) recruitment of eligible participants from the Harborne Ward of the Queen Elizabeth Hospital Birmingham, Mindelsohn Way, Birmingham, United Kingdom (UK); (2) baseline assessment of the secondary dependent variables related to multidimensional health; (3) assessment of the feasibility of the study as it relates to the eight-primary areas of focus for feasibility studies (acceptability, demand, implementation, practicality, adaptation, integration, expansion and limited-efficacy testing)[22]; (4) post-intervention assessment of all primary and secondary dependent variables.

The research questions of this study relate to the eight aforementioned areas of focus of this feasibility study, incorporating the following questions relating to the feasibility and efficacy of the study within this setting: Can it work? Will it work? Does it work?[22] (table 1).

### Design overview
This feasibility study will use a 2-week, interventional, independent measures research design (figure 1).

A variation of a stepped-wedged design/rolling recruitment will be utilised, with the interventions being conducted multiple times over the course of several months in order to maximise the potential sample size due to constraints of the setting: a delayed transfer of care hospital ward for patients prior to official discharge, with the majority of patients residing on the ward for >3 weeks, and ~25 patients on the ward at any given time.

The independent variables of the study will comprise of two exercise training interventions: a specially adapted machine-based resistance training intervention, and Move It Or Lose It (MIOLI), an established community-based exercise intervention for older adults. A control group will be utilised within the proposed future clinical trial but will not be utilised within this feasibility study as one of the primary purposes of this study is to assess the feasibility, and to a limited degree the efficaciousness, of the interventions within this setting (figure 2).

In order to ensure this present study is as scientifically valid as possible a number of precautions have been taken to protect the internal and external validity of the study within its methodological design: first, for each participant, all testing procedures (baseline and post-intervention (2 weeks)) will be conducted at approximately the same time of day (±2 hours). This will be controlled in order to protect the findings of the study from changes in the dependent variables which may be attributable to circadian variation rather

**Table 1** The eight primary areas of focus, outlining the research questions and methods of assessment

| Area of focus | Potential questions | Methods of assessment |
|---|---|---|
| Acceptability | ► Will the proposed population be interested in participating in the study?<br>► What will the uptake be?<br>► Will the programme be judged as suitable by the delivers of the programme in addition to the programme participants?<br>► Participant's opinions on hypothetically being randomised into a control group during a proposed future clinical trial?* | ► Participant uptake analysis (All participants approached and eligible for the study, all of those successfully recruited to the study).<br>► Semi-structured interviews with participants.<br>► Focus groups with study support staff. |
| Demand | ► Will the proposed population of hospital inpatients participate in the study?<br>► What will adherence rates be?<br>► Are the staff on the ward open to the idea of having exercise interventions potentially on the ward long term if it proves effective? | ► Analysis of uptake rates.<br>► Exercise intervention adherence rates.<br>► Focus groups with study support staff/ward staff. |
| Implementation | ► What are the possible logistical issues with the setting which will need to be addressed or accounted for prior to the clinical trial?<br>► Can the interventions be successfully carried out within this setting?<br>► Can a single or double bind be successfully implemented within this setting? | ► Semi-structured interviews with study participants.<br>► More in-depth with focus groups with study support staff. |
| Practicality | ► What are the practical implications of the study with relation to time commitment of the researchers, relating to both the implementation of the interventions, and the testing of participants for the dependent variables of the proposed future clinical trial?<br>► Is it viable to potentially conduct follow-up testing on participants in the proposed future clinical trial?<br>► Do any alterations need to be made to the proposed primary dependent variables of the future clinical trial?<br>► If the interventions are successful in influencing parameters of health and functional capacity, will it potentially be possible to assess if these improvements are sustained during a 2-week follow-up in the proposed future clinical trial, if the same is found? | ► Semi-structured interviews with study participants.<br>► Focus groups with support staff. |
| Integration | ► How will the ward staff appraise the study?<br>► Will the interventions be easily integrated into the existing culture, protocols and procedures within the ward seamlessly? | ► Focus groups with ward/support staff. |
| Adaptation | ► Will any further adaptations be required to the existing interventions to make them more feasible or appropriate within this setting? | ► Semi-structured interviews with participants. |
| Expansion | ► Can the Move It or Lose It intervention (an established chair-based exercise programme for older adults) be successfully expanded to this setting?<br>► Can be specially adapted resistance training equipment be successfully expanded to this setting? | ► Semi-structured interviews / focus group with the ward/study support staff. |
| Limited-efficacy testing | ► Is 2 weeks a sufficient duration to potentially provide significant benefit to patients?<br>► Can intensive (5–6 days/week), short duration (2 weeks) physical activity interventions improve markers of multidimensional health, in very frail individuals? | ► Analysis of the secondary dependent variables within the study (primary dependant variables of the future clinical trial).<br>► Analysis of uptake and adherence rates.<br>► Analysis of the level of satisfaction with the interventions through questionnaires with participants post-intervention. |

*Participants within the feasibility study will not be recruited as participants within the proposed future clinical trial in order to protect the scientific validity of the proposed future clinical trial, as the participants within the feasibility study will already have undergone the interventions, or similar interventions. However, participants within the feasibility study will likely not still be residents on the ward, as the average stay for participants on the ward is ~3.5 weeks. As such once the proposed future clinical trial has commenced, all participants within the feasibility trial will likely have left the ward well in advance.

than manipulation of the independent variables.[29] The hypothesis of the study will not be divulged to participants prior to or during the conduction of the study in order to control for any potential degree of demand characteristics; a scenario where participants alter their behaviour and/or answers, in order to align with what they believe is potentially the 'desired' outcome of the study. All dependent variable testing sessions will take place at least 24 hours after the cessation of the previous training session for each participant. This will be implemented to ensure acute fatigue does not become a contributing factor to the results of the study, specifically relating to the secondary dependent variables, but also the feasibility of such practice during a proposed future clinical trial. The order in which dependent variables are tested will be counter-balanced throughout the study at each assessment timepoint in order to attempt to protect the study from practice effects, especially order effects, where a

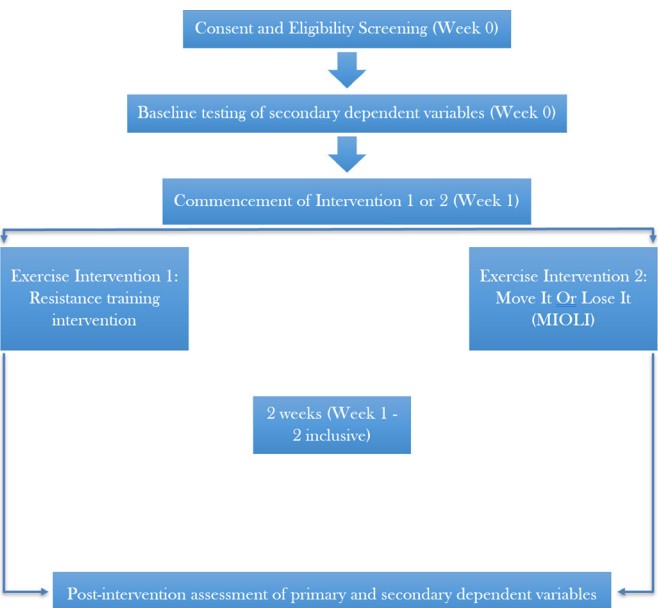

**Figure 1** Trial schema of participant flow throughout the duration of the study.

participant has been exposed to a specific order of testing before and as such performs better on subsequent testing procedures of the same material. Additionally, only one intervention will be run at a given time, protecting against potential subconscious selection bias among the research team relating to group allocation of participants. Finally, in order to increase the external validity of the study, eligibility criteria will be kept as minimalistic as possible (within the limits of safety and reason), in order to allow as inclusive a proportion of this population as possible, and in such producing findings which are applicable to not only those within the study, but to the greater population of frail geriatric hospital inpatients and particularly those within delayed transfer of care settings.

### Eligibility
This study is open to both men and woman whom meet the following eligibility criteria: inpatient on the Harborne ward of the Queen Elizabeth hospital Birmingham, Mindelsohn Way, Birmingham, UK; ≥65 years of age; frail according to the Fried Frailty Phenotype criteria[2]; ability to speak and read in English; not currently taking part in any other clinical trial which could potentially impact on or influence the findings of this study; not currently terminally ill with life expectancy less than the duration

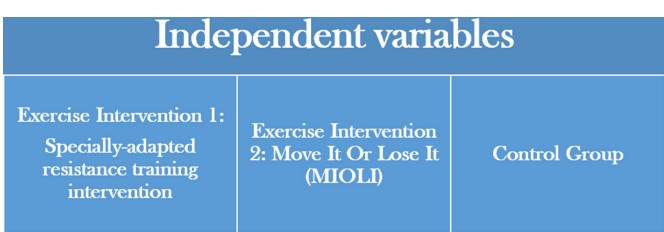

**Figure 2** The independent variables of the proposed future clinical trial.

of the study; no severe sensory impairment which would profoundly impact on ability to undergo the intervention, even once appropriate adaptations have been made; anticipated by their care team to remain on the ward for ~21 days post-enrolment into the study.

### Interventions
All participants will undergo 10 sessions in total throughout the 2-week intervention in either one of the two interventions. All session will be conducted individually under the guidance of a qualified trainer. A maximum of one session will be performed each day, and sessions will not be performed on any more than a maximum of three consecutive days throughout the duration of the study (table 2).

#### Exercise intervention 1: specially adapted resistance training intervention
This intervention will be comprised of an intensive (5 days/week), short duration (2 weeks), ~35 min per session, machine-based resistance training intervention. The exercises performed will specifically target the lower limbs through a combination of multijoint strength and power training utilising a leg press and leg extension machine (figure 3).

The maximal strength reference value (% one repetition maximum (1RM)), duration (35 min), type of exercise (multi-joint), loadings (60% 1RM (power), 80% 1RM (strength)) and volume (3 sets, 5–8 repetitions) are largely consistent with the position statement from the National Strength and Conditioning Association regarding resistance training in older adults.[30]

An outline of the protocol for each session can be found in figure 4.

#### Exercise intervention 2: MIOLI
This multi-component intervention will comprise of an intensive (5 days/week), short duration (2 weeks), ~35 min per session, chair-based exercise intervention. The exercises conducted will relate to strength, power, flexibility and aerobic capacity, consistent with the established MIOLI flexibility, aerobic capacity, balance and strength training programme for community-dwelling older adults. The exercises performed will predominantly targeting the lower limbs, but also incorporate the upper body and core.

An outline of the protocol for each session can be found in figure 5.

### Dependent variables
#### Primary dependent variables
The primary dependent variables will relate to the eight primary areas of focus of feasibility studies[22] (utilised to establish the feasibility of a proposed future clinical trial within this setting), relating to: acceptability, demand, implementation, practicality, adaptation, integration, expansion and limited-efficacy testing.

These dependent variables will be assessed through semi-structured interviews with study participants and

**Table 2** Study timeline of all major events during each round of recruitment (SPIRIT schedule)

| Week | Seated physical activity in ageing (SPAA) study timeline | | | | | | |
|---|---|---|---|---|---|---|---|
| | **Monday** | **Tuesday** | **Wednesday** | **Thursday** | **Friday** | **Saturday** | **Sunday** |
| 0 | Identification + participant information sheet distribution | 24 hours consideration period | Recruitment, eligibility screening and baseline assessments | Recruitment, eligibility screening and baseline assessments | Recruitment, eligibility screening and baseline assessments | Recruitment, eligibility screening and baseline assessments | – |
| 1 | Training | Rest | Training | Training | Rest | Training | Training |
| 2 | Training | Rest | Training | Training | Training | Rest | Training |
| 3 | Post-intervention assessment | Post-intervention assessment | Post-intervention assessment | Post-intervention assessment | Post-intervention assessment | Post-intervention assessment | – |

SPIRIT, Standard Protocol Items: Recommendations for Interventional Trials.

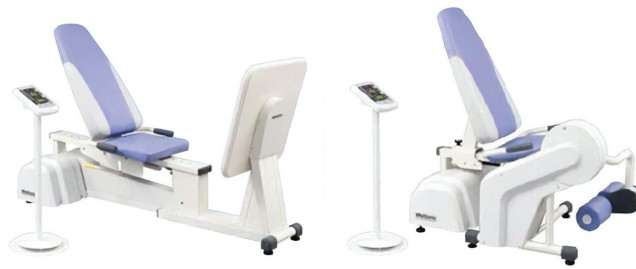

**Figure 3** Minato resistance training equipment utilised within the specially adapted resistance training intervention.

focus groups with ward staff post-intervention. Participant uptake and adherence records will also be employed throughout. These methods seek to attain answers to the following questions and parameters relating to the eight primary areas of enquiry for this feasibility study outlined in further detail in table 1.

In order to enhance trustworthiness in the qualitative component of this research, several methods will be employed:

The researcher gathering the data will keep a reflective journal in which they will record information about themselves, their activities and the methods used. Field notes will include time, date and location, participant's actual

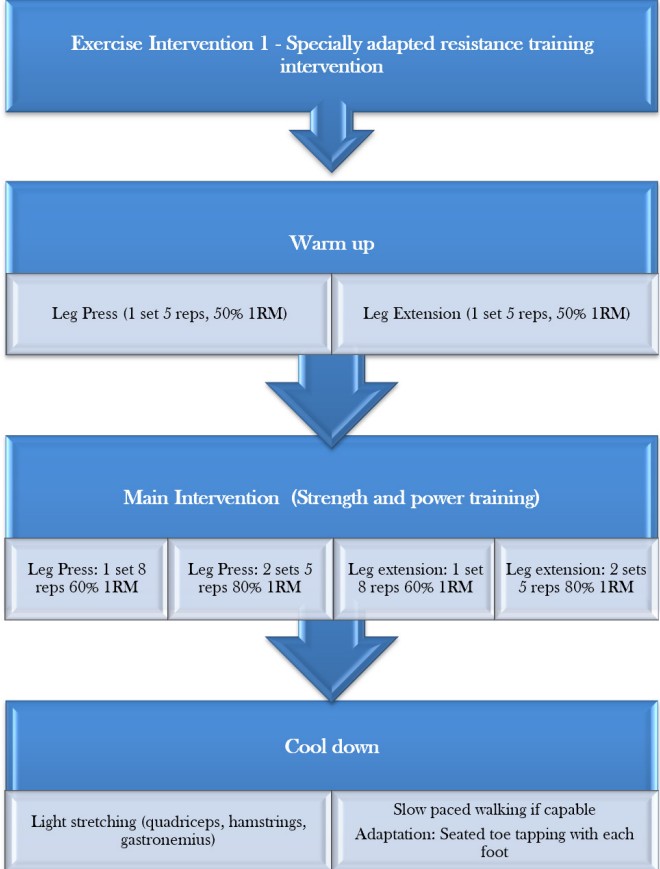

**Figure 4** Exercise intervention 1—specially adapted resistance training intervention protocol. 1RM, one repetition maximum.

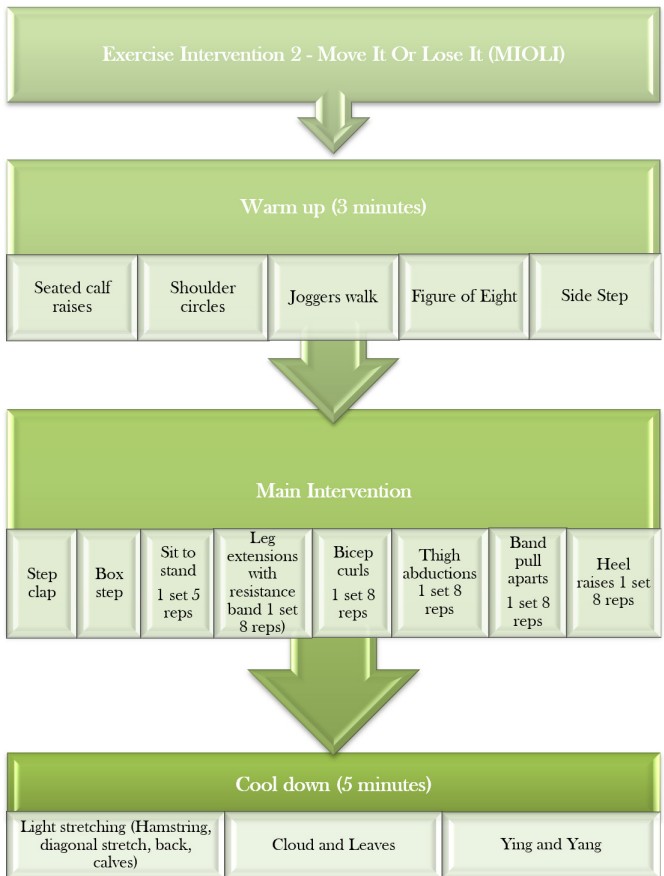

**Figure 5** Exercise intervention 2—Move It Or Lose It (MIOLI) chair-based exercise intervention protocol.

notes, and the researcher's own questions and comments. This will lend to logging and documenting what is learnt about the study, the intervention, the setting, the participants, and used to refine focus for future interviews through assessing the following questions: What is important? What is it I need to find out more about? What would I want to focus on more closely if I could do the interview again, or in future interviews?[31]

Data will be gathered from study participants and ward staff in order to collect data from multiple sources (triangulate information). This study will also employ more than one researcher to analyse the qualitative data in order to enhance triangulation and validity.

Enough details will be given about the participants and the setting to make decisions about the quality of the findings from the qualitative analysis. Detailed descriptions about the participants experiences and the setting will be provided by the researcher.

In the qualitative data analysis, clarification of all possible researcher biases will be made known. For example, it will be articulated that the researcher is an advocate of physical activity as a means to promote health, prescribing to the theoretical and practical concept of exercise as medicine, and hence there may be some form of unconscious subjective bias in this context. However, it should also be noted that the researcher within this study is also an advocate of science to an equal or even greater

extent, and as such any such bias in subjective analysis would potentially be counteracted in this sense.

Interviews—'a conversation with a purpose'[32] will be the primary method of data gathering utilised, as it enables large amounts of information to be gathered relatively quickly. Specifically, this study will employ semi-structured interviews, with open questions in a conversational format. There will be several predetermined themes, topics and questions to be discussed, specifically relating to the eight areas of focus of the primary dependent variables of this study. All interviews and focus groups will be audio recorded in order to facilitate future transcription. The qualitative element of this study will also explore opportunities for Patient and Public Involvement (PPI) in the research design of the proposed future clinical trial.

As this study will use a mixed-methods research approach, employing both qualitative and quantitative research methodologies, this will facilitate the potential for elaboration and expansion of these findings of individual methodologies through complementary analysis. The qualitative aspect of this feasibility study, aimed at assessing the primary dependent variables, will predominantly take a phenomenological approach to understand the experiences of individuals involved in the study.[33]

## Secondary dependent variables
The secondary dependent variables relating to multidimensional health (comprising the proposed primary dependent variables of the future clinical trial) will be:

Physiological*: serum cortisol, dehydroepiandrosterone-sulphate (DHEAS), cortisol: DHEAS ratio, C reactive protein (CRP), IL-6, tumour necrosis factor alpha (TNFα), interferon gamma (IFNγ).

Functional: hand grip strength (Southampton protocol,[34] Leg strength and power output,[35 36] Short Physical Performance Battery (SPPB),[37] Katz Index of Independence in Activities of Daily Living (Katz ADL),[38] and the Fried Frailty Phenotype)[2]

Psychological/emotional: Geriatric Depression Scale,[39] Hospital Anxiety Depression Scale.[40]

Cognitive: Standardised Mini-Mental State Examination.[41]

Social: Interpersonal Support Evaluation List (ISEL-12).[42]

*All blood samples will be obtained through venepuncture. Serum will be analysed for the physiological dependent variables relating to cortisol, DHEAS and CRP (assessed by commercial ELISA kit). Inflammatory cytokines (IL-6, TNFα, IFNγ) will be assayed using a multiplex commercial kit (R&D Systems). These specific physiological variables will be examined due to their association with the ageing process, and previous research which have both proposed and indicated that exercise is potentially capable of altering these variables.[43–48]

## Data collection
Data will predominantly be collected at two main time points: baseline and post-intervention (table 2).

## Baseline assessment

Participants' baseline sociodemographic and information for the secondary dependent variables will be collected between 5 and 2 days prior to intervention commencement. 1RM for the specially adapted resistance training equipment will also be assessed during this time period (after all baseline testing has been completed, and at least 24 hours after baseline testing which requires physical exertion, which may impact on the accuracy of the 1RM measurements). 1RM will be determined based on estimation from participants five repetition maximum (5RM) utilising the Epley formula: $1RM = w (1 + r/30)$, where 1=weight, and r=repetitions.[36] The protocol employed during obtainment of participants 5RM has been adapted from[35] (online supplementary file 1). A balance screening will also be conducted prior to the commencement of the MIOLI intervention to determine whether participants should perform exercises standing, standing with chair support, or seated. In addition, the resistance bands which participants in the MIOLI intervention will use will similarly be assessed during this period; with three options corresponding to light, moderate and high resistances, prescribed to participants based on their initial ability and preference during their performance of the exercises in which the resistance bands will be utilised.

## Post-intervention assessment

Primary and secondary dependent variables will be assessed between 1 and 5 days post-intervention cessation. All assessments will be conducted at least 24 hours post the cessation of the last training session.

Adherence rates in the intervention will be recorded as the number of repetitions completed in a set (100% required for adherence to that exercise), and the number of exercises for which there was 100% adherence. If participants meet these parameters for each exercise session, they will be considered to be in 100% adherence to the intervention. For example, if a participant adheres to 100% of the intervention for nine sessions, but only 90% for one session, then they will have a 99% adherence rate. Adherence rates, whether high, or low, may signify that the interventions may have been too demanding, too easy or optimal. Information will also be collected throughout the study related to uptake and retention rates.

## Data monitoring

Data will be monitored by the trial management committee at monthly intervals. Prior to analysis data entry checking will be conducted for accuracy on 10% of all participants, and queries resolved through discussion with the trial management committee and access to the source documents. Data management will adhere to the Physical Activity and Nutritional INfluences In ageing (PANINI) project data management plan, which was developed in accordance with national and European principles as part of the university research governance and European Commission research governance guidelines. Thus, data management for this project adheres

to the Findable, Accessible, Interoperable, and Reusable (FAIR) principles.[49]

## Sample size

This study aims to recruit a convenience sample of n =~30 participants: 15 in each intervention. No formal power calculations were conducted due to the feasibility nature of this study. This sample size was based on initial expectations related to the obtainment of data saturation in the qualitative component of the study given its phenomenological approach.[50 51]

## Identification, consent and recruitment
### Identification

Patients on the ward at the commencement of the study will initially be screened by their care team for the known presence of any severe sensory impairments which would exclude them from participation. Due to the study's rolling recruitment, this initial screening will continue throughout the study as additional patients are admitted to the ward. Participant information sheets will not be distributed to patients who are deemed to be medically unfit or due for imminent discharge as identified by their care team. All other aspects of the screening process will occur after consent has been obtained. After identification, potential participants will be approached by the researcher with an information sheet related to the study and asked if they would be interested in participating, or if they would like to receive more information about the study (online supplementary file 2). The information sheet will contain all the most pertinent information relating to the study and specifically what it would require from potential participants. Potential participants will be given ≥24 hours to consider whether or not they would like to participate.

### Consent

At this stage, potential participants will also be provided with an informed consent form and asked if they would be interested in participating (online supplementary file 3). If it is deemed that a potential participant lacks the capacity to consent, a personal consultee will be sought. A personal consultee will be someone who cares for the patient, or is interested in their welfare (but not a paid professional), and who is prepared to be consulted and give advice on what they believe the patient's wishes would be, were they to have the capacity to consent for themselves. If a personal consultee cannot be found, a nominated consultee will be sought. A nominated consultee will be someone who is familiar with the patient in a professional context and can adequately advise on whether they believe participation would be in the patients' best interest. These processes will be informed by the UK Department of Health's guidance on nominating a consultee for research involving adults who lack the capacity to consent[52]; in accordance with section 32[3] of the Mental Capacity Act 2005.[53] All efforts possible will be made in this regard to include participants whom lack

the capacity to consent, as intrinsically within the research team from a personal and professional perspective we would consider it unethical to exclude potential participants from participating in a study, which can potentially benefit them and their overall health status, due to the fact that they lack the capacity to consent.

### Recruitment
Following consent being obtained from the participant themselves, or the obtainment of a declaration from a consultee, all consented participants will be screened for the remaining eligibility criteria.

### Statistical methods
#### Primary dependent variables
Analysis of the primary dependent variables will be based on an inductive process utilising Interpretative Phenomenological Analysis (thematic analysis). Two researchers will be employed to analyse the data to increase triangulation from the analysis perspective, having already triangulated data acquisition through obtainment from multiple sources (study participants and ward staff). All semi-structured interviews and focus groups will be audio recorded. Data synthesis will be performed through verbatim transcription. The three main steps of Interpretative Phenomenological Analysis will be followed[33]: (1) The generation of themes from transcripts within the areas of feasibility inquiry. As an iterative process, these themes will be continuously reviewed and adapted based on the emergence of information in subsequent transcripts. (2) The collation and separation of these themes within each of the areas of feasibility inquiry. (3) Written interpretation of the resultant themes within each of the areas of feasibility and their relationship to one another. At all stages within this process, reflective journal entries and field notes will be utilised to provide a more comprehensive understanding of the findings, in addition to incorporating additional feasibility information related to uptake and retention rates, and limited efficacy testing of the secondary dependent variables in the final analysis to provide a comprehensive assessment of the feasibility of the study.

#### Secondary dependent variables
Statistical analysis of the secondary dependent variables will be performed using IBM Statistical Package for Social Sciences (SPSS) software. These analyses will be performed as part of the limited efficacy testing regarding the potential impact of the interventions on the secondary dependent variables (proposed primary dependent variables of the future clinical trial). These analyse will provide an estimation of efficacy and provide valuable insight regarding feasibility; useful to informing the design of the future powered clinical trial. Specifically, the following statistical analysis will be utilised: 2×2 way independent measures analysis' of variance consisting of two independent variables; the specially adapted resistance training intervention, and the MIOLI intervention, each with two

levels: baseline and post-intervention. A subsequent post hoc test will be utilised if a significant main effect or interactions are found. Pearson product correlations will also be utilised between various sociodemographic variables (such as age and sex) and the dependent variables of this study, to assess possible relationships between differences in sociodemographic factors and changes in the dependent variables.

Central tendency and variability measurements consisting of parameters such as the mean, median and mode, and SD and range of scores, respectively, will also be utilised during the analysis of data for illustrative purpose. Significance levels will be set at 0.05 ($p \leq 0.05$), and effect sizes will be reported for all analyses. In order to establish if the assumptions of parametric statistics have been met regarding the assumption that there is a normal distribution of data, the data will be analysed for skewness and kurtosis. In the event, data do not fulfil the assumptions of parametric statistics, the non-parametric equivalent of the aforementioned statistical analysis will be employed, namely the Scheirer-Ray-Hare test, and Spearman's rank-order correlations. As the quantitative component of this study has not been powered given the feasibility nature of the study, the examination of the efficacy of the intervention to impact these secondary dependent variables is limited; and interpretation of these results should be treated with caution pending the future powered clinical trial. All results will be reported with effect size and 95% CIs.

### Data storage and protection
Participants' identity or other personal information will be kept confidential. Participants will be assigned a unique ID number under which all study information will be stored in a secure file on an encrypted and password-protected computer and laptop at the University of Birmingham (UoB). Physical data (eg, Case Report Forms (CRFs)) will be identifiable only by ID number and stored in a locked filing cabinet at the School of Sport, Exercise and Rehabilitation Sciences at UoB, accessible only by the research team. Participants' personal data (name, date of birth) and consent forms matching them to their ID number, will be stored securely in a locked filing cabinet, separate from all other data and/or in a password-protected master sheet on an encrypted and password-protected computer and laptop at UoB. All serum samples will be stored in Human Tissue Act compliant facilities at UoB for 3 years then destroyed.

All hard copy data collected on CRF's will be stored in a linked-anonymised format securely for 10 years then destroyed. All personal data (consent forms, master sheet linking participant IDs to names and contact details) will be stored for 10 years then destroyed. All computerised data will be archived on UoB servers in anonymised form for 10 years in the first instance in accordance with the UoB Code of Practice for Research, and the Data Protection Act (2018).

Following analysis for this specific study, all data will be anonymised and entered into a European 'PANINI' open access database that this project is part of and may be analysed in future ethically approved research across the PANINI network. The PANINI shared dataset will be made open access at the conclusion of the funding for the PANINI network including this study in 2020 and stored for at least 10 years as an open access searchable published dataset.

## Patient and public involvement

All authors are strong proponents of PPI and engagement with research and believe the findings of the study will be important to aid the facilitation of improvements in the care of frail older hospital inpatients. Given the feasibility nature of this research, the qualitative element of this study will explore opportunities for PPI in the research design of the future clinical trial. The findings of this study will be disseminated to participants on request, and our PPI groups.

## ETHICS AND DISSEMINATION
### Dissemination

The findings of this study will be disseminated through publication of scientific papers in open access scientific journals, public engagement events, online via social media (Twitter, Instagram) and the PANINI project website,[54 55] presentation at various conferences, and to study participants on request.

## Safety reporting and monitoring

Adverse events (AEs) and serious adverse events (SAEs) will be monitored and if applicable recorded, by the primary researcher at each testing or exercise session and reported weekly to the principle investigator (PI) and reviewed in the case of AEs. SAEs will be reported immediately to the PI who will complete an SAE form indicating causality and severity. The chief investigator (CI) will then submit this to the sponsor's research governance office, the Research Ethics Committee and University Hospitals Birmingham (UHB) research governance office, within 24 hours. UHB local policies and Standard Operating Procedures for all safety reporting will also be followed by the research team. SAEs related to pre-existing conditions will not be reported. Actions following an AE or SAE would be as standard, that is, direct referral to their clinical care team, who then may wish to treat the AE or SAE themselves or where appropriate refer the participant to another relevant medical professional, and to recommend that the participant withdraw from the study unless they have been cleared to continue exercise by their attending physician.

## Trial registration

This study has been registered on ClinicalTrials.gov under the identifier number: NCT03141866.

## Trial status

This trial commenced on the 3 September 2018, on the Harborne ward of the Queen Elizabeth Hospital Birmingham, with anticipated completion of data collection scheduled for 9 August 2019. This present study is a sister trail of another feasibility study; comprising of a specially adapted resistance training intervention for frail geriatric care home residents.[56]

**Author affiliations**
[1]School of Sport, Exercise and Rehabilitation Sciences, University of Birmingham, Birmingham, UK
[2]MRC-Arthritis Research UK Centre for Musculoskeletal Ageing Research, Institute of Inflammation and Ageing, University of Birmingham, Birmingham, UK
[3]NIHR Birmingham Biomedical Research Centre, University Hospitals Birmingham NHS Foundation Trust and the University of Birmingham, Birmingham, UK
[4]Faculty of Health Sciences and Sport, University of Stirling, Stirling, UK

**Contributors** PD designed the study protocol, and associated manuscript for publication, with supervision, input and feedback from ACW at all stage of the design and writing process. JML and CAG reviewed and revised the manuscript prior to publication. All authors have read and approved the final manuscript.

**Funding** This review has been funded by the European Commission's Horizon 2020 research and innovation programme under the Marie Sklodowska-Curie grant agreement (675003); of which PD is a Marie Sklodowska-Curie Doctoral Research Fellow, ACW, JML and CAG doctoral supervisors, and ACW the grants Principal Investigator.

**Competing interests** None declared.

**Patient consent for publication** Not required.

**Ethics approval** This study has been granted a favourable ethical opinion by the Coventry and Warwickshire NHS Research Ethics Committee (REC) (17/WM/0390) on the 12/03/2018. This study has also been sponsored by the University of Birmingham, after review by the sponsor's research governance office (sponsor registration number: RG_17–108).

**Provenance and peer review** Not commissioned; externally peer reviewed.

**ORCID iD**
Paul Doody http://orcid.org/0000-0001-6732-1384

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
