## [Reviewer comments · BMJ Open]

ARTICLE DETAILS

TITLE (PROVISIONAL)	Assessing the feasibility and impact of specially adapted exercise interventions, aimed at improving the multi-dimensional health and functional capacity of frail geriatric hospital inpatients: protocol for a feasibility study.
AUTHORS	Doody, Paul; Lord, Janet; Greig, Carolyn; Whittaker, Anna

VERSION 1 – REVIEW

REVIEWER	Freiberger Ellen Friedrich-Alexander-University Erlangen-Nürnberg Institute for Biomedicine of Aging Germany
REVIEW RETURNED	30-May-2019

GENERAL COMMENTS	The authors present the study protocol of a feasibility study in geriatric frail older persons in the hospital setting. The authors want to test the feasibility and efficacy of an exercise intervention with a duration of two weeks in the hospital setting. Although this is an important topic and the manuscript includes a study protocol, major comments remain before the protocol can be considered for publication: Major comments: Introduction: As the authors rightfully stated that well planned exercise intervention in frail geriatric patients are needed in their introduction they rarely address the need of an exercise intervention during the hospital stay. The reviewer would suggest to add information on percentage of hospital stays in the frail older population; information in time in bed during the hospital (see Theou et al) and the average stay of frail older persons in the hospital setting. Right now, the introduction leads nicely into the topic of frailty but not into the needs of intervention of frail geriatric patients in the hospital setting. Especially information on length of hospital stay will be needed as later on the authors stated 3 weeks. Methods: The authors stated a holistic approach in the future main trial but there is no primary goal defined. The reviewer wonders if the stated effects “on the physiological, psychological, 103 cognitive, social, and emotional health, and functional capacity of frail geriatric populations” (p 5 line 23-27) will all be the primary outcome. Even in the feasibility study the future main goal should be stated and not be defined based on the results of the feasibility study.
---

	Furthermore, the authors stated that all will be related to the 8 aforementioned areas (p5 line 43-45) but the reviewer think at least the 8 presented topics in Table 1 should be verbally mentioned at this point. For the reviewer no arguments are given that the MIOLI intervention can be transferred to the hospital setting. Please provide a rational for this approach. The authors are to congratulate for their well-planned methods of validations. It demonstrate the high expertise of the research group. Out of curiosity, the reviewer wonders if there is cognitive exclusion cut-off, and how the research group will rule out patients with delir. Please just comment. The planned time of the exercise intervention are not supported by the introduction. Please provide information, why the authors aim at 35 minutes per session. With regard to the qualitative part of the research, the authors are again to congratulate for their thoughtful design. With regard to the secondary objective measures (p 10 line 50-60 ff) the reviewer wonders –out of own experience – if the SPPB will be feasible in the frail geriatric patients. Furthermore, the authors should state on how they want to obtain “One repetition maximum (1RM) for the specially adapted resistance training equipment” in a frail geriatric hospital patient (p 11 line 46-48). This does seem very optimistic for the reviewer. With regard to obtaining the IC the process described seems very vague, and due to the EU data safety law the reviewer wonders if this would be appropriate. In the country of the reviewer you would need a legal carer for obtaining the IC, if the patient is not able or fit to provide the IC: Please add a section of strength and limitation to the study, elaborating on possible barriers, and how to solve the barriers. Minor comments: With regard to the key words the reviewer would suggest to delete elderly (as older adult is already in) but relate more to the short term nature of the intervention.
--	---

REVIEWER	Edu Carballeira Gerontology Research Group Department of Physiotherapy, Medicine & Biomedical Sciences. University of A Coruña INIBIC- University Hospital Complex A Coruña (CHUAC) Faculty of Health Sciences, Campus de Oza, 15006, A Coruña Spain
REVIEW RETURNED	05-Jun-2019

GENERAL COMMENTS	The present study protocol will assess the feasibility and efficacy of short duration (2-week), intensive (five-days per week) specially adapted exercise interventions within a delayed transfer of care hospital ward setting. I firmly believe that the aim of the study is a interesting topic in the area of health intervention in the geriatric population. Nevertheless, I have some concerns about the design of the study the should be addressed. General comments...
--

	The main concern lies in the definition of the independent variable. The objective of the study is assess the feasibility and efficacy of two exercise protocol, one is an adapted machine-based resistance training intervention, and the other one a chair-based exercise intervention (Move It Or Lose It, MIOLI). Both protocols will last 35 min, however, duration is one of the multiple variables that defines training programme, and when a comparison between two exercise protocols is going to be made, the authors should define the differentiating variable (p.e. volume of muscles implicated) and should explain how they are going to equate the rest of the variables (p.e. type of exercise, intensity, volume, density, recovery,...). There are many evidence that different combinations of the variables that defines training programme, the dose, influence on dependent variables, the response. That dose-response has to be well defined in order to know with some certainty what variable of the exercise protocol produced the outcomes observed in the dependent variables (physiological, functional, emotional, psychological / emotional, cognitive and social). For these reasons, I encourage the authors to apply two exercise protocols where the variable to be studied is well controlled, avoiding the confounding variables. Specific comments... Abstract_ Methods and analysis: details about independent variable should be provided. Background_ _page 4, line 5_ Paraphrase the sentence, as is redundant. _page 6, line 22_ To state the differentiating variable between the two protocols. Methods and analysis_ _page 8, line 26 and figure 4_ are te authors sure that repetition maximum is the best choice to evaluate the strength on geriatric hospitalized patients? Maybe other strength evaluations and monitoring tools could be employed, as maximum number of repetitions, velocity-based evaluations and velocity-based trainings, monitoring throughout perception of effort scales (i.e. OMNI-RPE monitoring). I encourage the authors to evaluate other options more suitable for this population, since the realization of the protocols of 1RM may lack reliability and may even be dangerous in this population. _page 10, line 50_ why these physiological variables are going to be analyzed? please explain your selection. Statistical methods_ Secondary dependent variables_ page 14, line 48_ the authors must explain what analyzes would be carried out in case of outcomes do not fulfill the assumptions of normality.
--	--

VERSION 1 – AUTHOR RESPONSE

Reviewer(s)' Comments to Author:

Reviewer: 1

Reviewer Name: Freiberger Ellen

Institution and Country: Friedrich-Alexander-University Erlangen-Nürnberg

Institute for Biomedicine of Aging
Germany

Please state any competing interests or state 'None declared': None declared

Please leave your comments for the authors below

Comment - The authors present the study protocol of a feasibility study in geriatric frail older persons in the hospital setting. The authors want to test the feasibility and efficacy of an exercise intervention with a duration of two weeks in the hospital setting. Although this is an important topic and the manuscript includes a study protocol, major comments remain before the protocol can be considered for publication:

Response - Dear Dr. Freiburger, Thank you for your review of our manuscript. Please find a response to all of your comments addressed below, and alterations highlighted within the manuscript as red text.

Comment - Major comments:

Introduction:

As the authors rightfully stated that well planned exercise intervention in frail geriatric patients are needed in their introduction they rarely address the need of an exercise intervention during the hospital stay. The reviewer would suggest to add information on percentage of hospital stays in the frail older population; information in time in bed during the hospital (see Theou et al) and the average stay of frail older persons in the hospital setting. Right now, the introduction leads nicely into the topic of frailty but not into the needs of intervention of frail geriatric patients in the hospital setting. Especially information on length of hospital stay will be needed as later on the authors stated 3 weeks.

Response - The authors have now added additional information to the introduction to address the need for exercise interventions during hospital stay on Lines 87 – 96 (added text in bold):

*“Moreover acute hospital admission for older adults are associated with further loss of physical activity and represent a period of increased susceptibility to sarcopenia and frailty (29). **Frailty is associated with longer stay and increased rates of mortality in hospitalised older adults, as well as serving as a predictor of readmission (12, 30).** Therefore, there is an urgent need to examine the effect of such interventions within this setting, and whether these interventions can be employed to improve various aspects of health in frail older populations in inpatient hospital ward settings, as well as their efficacy in specifically treating, preventing and reversing frailty. **Preliminary research has shown some success in the implementation of exercise interventions to reverse functional decline in general geriatric inpatient populations (31), however, to the authors’ knowledge this present study is the first to attempt such interventions in frail delayed transfer of care patients.**”*

The greater than three weeks length of stay is specific to the ward the study will take place on, as referenced on lines 121 - 123:

“a delayed transfer of care hospital ward for patients prior to official discharge, with the majority of patients residing on the ward for > 3 weeks, and approximately 25 patients on the ward at any given time.”

Greater than 3 weeks was the average length of stay we were provided with to then design the study around. Hence, it’s inclusion here when describing the specific setting with regard to its nature, average length of stay and bed capacity.

Added manuscript references:

12. Hao, Q., Zhou, L., Dong, B., Yang, M., Dong, B. & Weil, Y. 2019, "The role of frailty in predicting mortality and readmission in older adults in acute care wards: a prospective study", *Scientific reports*, vol. 9, no. 1, pp. 1207.
30. Khandelwal, D., Goel, A., Kumar, U., Gulati, V., Narang, R. & Dey, A.B. 2012, "Frailty is associated with longer hospital stay and increased mortality in hospitalized older patients", *Journal of Nutrition, Health and Aging*, vol. 16, no. 8, pp. 732-735.
31. Martínez-Velilla, N., Casas-Herrero, A., Zambom-Ferraresi, F., de Asteasu, Mikel L Sáez, Lucia, A., Galbete, A., García-Baztán, A., Alonso-Renedo, J., González-Glaría, B. & Gonzalo-Lázaro, M. 2019, "Effect of exercise intervention on functional decline in very elderly patients during acute hospitalization: a randomized clinical trial", *JAMA internal medicine*, vol. 179, no. 1, pp. 28-36.

Comment - Methods:

The authors stated a holistic approach in the future main trial but there is no primary goal defined. The reviewer wonders if the stated effects "on the physiological, psychological, 103 cognitive, social, and emotional health, and functional capacity of frail geriatric populations" (p 5 line 23-27) will all be the primary outcome. Even in the feasibility study the future main goal should be stated and not be defined based on the results of the feasibility study.

Response - *The primary aim of the future clinical trial is stated on line 99 - 103 of the manuscript as follows:*

"The primary aim of this study is to assess the feasibility of a proposed future trial in this setting, which aims to assess the impact of specially adapted exercise interventions on the physiological, psychological, cognitive, social, and emotional health, and functional capacity of frail geriatric populations within a hospital ward setting; recognising health as a multi-factorial concept incorporating multiple inter-related dimensions."

We believe the reviewer may have initially misread this line. It states that the primary aim of this study is to assess the feasibility of the proposed future trial. It then states that the primary aim of the proposed future clinical trial is to "assess the impact of specially adapted exercise interventions on the physiological, psychological, cognitive, social, and emotional health, and functional capacity of frail geriatric populations within a hospital ward setting"

We have also further alluded to the primary aim of the future trial within added text to the manuscript on lines 337 - 339:

"These analyses will be performed as part of the limited efficacy testing regarding the potential impact of the interventions on the secondary dependent variables (proposed primary dependent variables of the future clinical trial)."

Comment - Furthermore, the authors stated that all will be related to the 8 aforementioned areas (p5 line 43-45) but the reviewer think at least the 8 presented topics in Table 1 should be verbally mentioned at this point.

Response – *The authors agree with the reviewer and have now amended as follows on Lines 109 - 110 (additive text in bold):*

*"3) Assessment of the feasibility of the study as it relates to the eight-primary areas of focus for feasibility studies (**acceptability, demand, implementation, practicality, adaptation, integration, expansion and limited-efficacy testing**) (25)."*

Comment - For the reviewer no arguments are given that the MIOLI intervention can be transferred to the hospital setting. Please provide a rationale for this approach.

Response – *The authors themselves are uncertain if the Move It Or Lose It intervention can be successfully transferred to the hospital setting, hence the feasibility nature of this study. This exact inquiry is actually an area of feasibility inquiry for the study with regard to the transferability (expansion) of the interventions as stated in Table 1 on Line 600 under the heading “Expansion” – “Can the Move It or Lose It intervention (an established chair-based exercise programme for older adults) be successfully expanded to this setting?”.*

A promising aspect of the MIOLI intervention (in comparison to the resistance training intervention) is that it can be adapted based on the functional ability of the patients, for example it can be performed standing independently, standing with chair support, seated, or even an adapted version in the hospital bed for bedbound patients; so is potentially promising with regard to its expansion to this setting given its relatively high degree of adaptability – all of which will be eventually addressed in the results paper.

Comment - The authors are to congratulate for their well-planned methods of validations. It demonstrates the high expertise of the research group.

Response – *The authors thank the Reviewer for her complimentary remarks.*

Comment - Out of curiosity, the reviewer wonders if there is cognitive exclusion cut-off, and how the research group will rule out patients with delir. Please just comment.

Response - *There is no specific cognitive exclusion cut off criteria for this study. Provided patients meet the eligibility criteria listed on lines 155 - 163 of the manuscript, they are eligible for participation within the study. Every effort is made to include patients which meet these criteria within the study, either through their own informed consent or a consultee declaration. In the case of patients with delirium, they are unlikely to be eligible for the study given the aforementioned eligibility criteria, particularly as it relates to:*

“no severe sensory impairment which would profoundly impact upon ability to undergo the intervention, even once appropriate adaptations have been made... ability to speak and read in English”

As such in addition to not possessing any impairments that would profoundly impact upon their ability to undergo the intervention, participants must also have the ability to understand, follow and engage in basic verbal and written communication. Delirious patients will likely be initially screened out during the initial screening by their care team.

Comment - The planned time of the exercise intervention are not supported by the introduction. Please provide information, why the authors aim at 35 minutes per session.

Response – *This is primarily a projected estimation based on the exercises to be performed, rather than a chosen time period; it is simply the time it will take to complete the exercises, though dual concern was given to 1. The load and volume participants were asked to perform during each session 2. The overall length of the session. In this regard consideration was given to providing an adequate stimulus to participants to facilitate adaptation, while also not being too demanding as to cause undue fatigue in these frail geriatric patients throughout the intervention, and to facilitate participant adherence and retention throughout the study. These considerations also took into account previous studies in frail geriatric populations that were of a less intense nature (two-three times per week), but*

reported low adherence rates, which may have been due to the extended length of the intervention sessions, some as long as 60 - 90 minutes.

We have now altered the manuscript as follows on Lines 172 - 176 and Lines 179 - 182 to clarify that this is an approximate time period based on the exercises to be performed (additive text in bold):

Lines 172 - 176:

*“This intervention will be comprised of an intensive (five days per week), short duration (2 week), **approximately 35 minutes per session, machine-based resistance training intervention. The exercises performed will specifically target the lower limbs through a combination of strength and power training utilising a leg press and leg extension machine (Figure 3). (Insert Figure 3)***

An outline of the protocol for each session can be found in Figure 4. (Insert Figure 4)”

Lines 179 - 182:

*“This intervention will comprise of an intensive (five days per week), short duration (2 week), **approximately 35 minutes per session, chair-based exercise intervention. The exercises conducted will relate to strength, power, flexibility and aerobic capacity, predominantly targeting the lower limbs, but also incorporating the upper body and core. An outline of the protocol for each session can be found in Figure 5. (Insert Figure 5)”***

Comment - With regard to the qualitative part of the research, the authors are again to congratulate for their thoughtful design.

Response – *The authors again thank the Reviewer for her complimentary remarks.*

Comment - With regard to the secondary objective measures (p 10 line 50-60 ff) the reviewer wonders –out of own experience – if the SPPB will be feasible in the frail geriatric patients. Furthermore, the authors should state on how they want to obtain “One repetition maximum (1RM) for the specially adapted resistance training equipment” in a frail geriatric hospital patient (p 11 line 46-48). This does seem very optimistic for the reviewer.

Response – *With regard to SPPB assessments, thus far the measurements have been feasible for some, while for others certain aspect are difficult or else not feasible, for instance the chair rise and balance components; all of which adds to informing the feasibility of proposed future clinical trial. With regard to one repetition maximum assessments (1RM), this is based on an estimate from participant’s five repetition maximum (5RM) measurement, and thus far has been successfully implemented in all patients within the resistance training intervention without issue. It is worth noting that participants are performing 5 repetitions at 80% of their estimated 1RM within the intervention; typically the maximal load on a 5RM assessment will reach no more than 85% of participants 1RM. Additionally it is worth noting that all loads lifted during the course of the study are relative to the participants ability, and 5RM assessments are quite feasible. Though the Reviewer is correct that a direct 1RM assessment of course is generally not appropriate, advisable or necessary in any population other than young healthy individuals, hence an estimated 1RM based on 5RM was chosen. The Epley formula is then employed to provide an estimate of participants 1RM. In addition to providing a measure of maximal strength to act as a dependent variable, this is also used to calculate the load participants will lift within the specially adapted resistance training intervention as outlined in Figure 4 i.e. 50%, 60% and 80% of participants estimated 1RM for the warm up, power, and strength components of the intervention respectively. We have now added additional information regarding this to the manuscript on lines 259 - 261, as well as a supplementary file outlining the 5RM assessment protocol (Supplementary file 1):*

"1RM will be determined based on estimation from participants five repetition maximum (5RM) utilising the Epley formula: $1RM = w(1 + r/30)$, where w = weight, and r = repetitions (38). The protocol employed during obtainment of participants 5RM has been adapted from (37) (Supplementary file 1).

Added manuscript references:

38. Epley, B. 1985, "Poundage chart", *Boyd Epley Workout*. Lincoln, NE: Body Enterprises, 2985, vol. 86, pp. p. 86.
37. G.G. & Triplett, N.T. 2015, *Essentials of strength training and conditioning 4th edition*, Human kinetics.

Comment - With regard to obtaining the IC the process described seems very vague, and due to the EU data safety law the reviewer wonders if this would be appropriate. In the country of the reviewer you would need a legal carer for obtaining the IC, if the patient is not able or fit to provide the IC:

Response - We have now amended this section on Lines 300 - 316 as follows to provide additional information on the consent process (additive text highlighted in bold):

"Consent: At this stage potential participants will also be provided with an informed consent form and asked if they would be interested in participating (**Supplementary file 3**). If it is deemed that a potential participant lacks the capacity to consent, a personal consultee will be sought. **A personal consultee will be someone who cares for the patient, or is interested in their welfare (but not a paid professional), and who is prepared to be consulted and give advice on what they believe the patient's wishes would be, were they to have the capacity to consent for themselves.** If a personal consultee cannot be found, a nominated consultee will be sought. **A nominated consultee will be someone who is familiar with the patient in a professional context and can adequately advise on whether they believe participation would be in the patients' best interest. These processes will be informed by the UK Department of Health's guidance on nominating a consultee for research involving adults who lack the capacity to consent (54); in accordance with section 32(3) of the Mental Capacity Act 2005 (55).** All efforts possible will be made in this regard to include participants whom lack the capacity to consent, as intrinsically within the research team from a personal and professional perspective we would consider it unethical to exclude potential participants from participating in a study, which can potentially benefit them and their overall health status, due to the fact that they lack the capacity to consent.

Recruitment: Following consent being obtained from the participant themselves, or the obtainment of a declaration from a consultee, all consented participants will be screened for the remaining eligibility criteria."

Added manuscript references

54. United Kingdom. Department of Health 2008, *Guidance on nominating a consultee for research involving adults who lack capacity to consent* (8953). URL: https://webarchive.nationalarchives.gov.uk/20130123193236/http://www.dh.gov.uk/en/Publicationandstatistics/Publications/PublicationsPolicyAndGuidance/DH_083131. Accessed: 5th July 2019.
55. United Kingdom Mental Capacity Act, 2005. URL: <https://www.legislation.gov.uk/ukpga/2005/9/section/32>. Accessed 5th July 2019.

Comment - Please add a section of strength and limitation to the study, elaborating on possible barriers, and how to solve the barriers.

Response – The strengths and limitation section of this manuscript can be found on Lines 38 - 44, directly subsequent to the abstract in bullet point format as per BMJ Open’s formatting requirements:

“Strengths and limitations

- *Mixed methods feasibility study employing both quantitative and qualitative research methodologies*
- *Specially adapted exercise interventions for frail geriatric populations*
- *Difficult to reach (and often excluded) participant population*
- *Single site study”*

Comment - Minor comments:

With regard to the key words the reviewer would suggest to delete elderly (as older adult is already in) but relate more to the short term nature of the intervention.

Response - *The authors would ideally prefer to retain the term elderly within the key words, as they believe that this term is more likely to bring increased attention to the manuscript than for example “short-duration”. Although not favoured term by some, it is a commonly utilised and accepted term in many parts of the world, denoting, or conferring, respect or politeness, and most importantly commonly utilised in research; most noticeably in North America. As such the authors believe its inclusion within the key words is more likely to bring increase exposure to the manuscript than if it were absent, even with the inclusion of the terms “geriatric” and “older adult”.*

Reviewer: 2

Reviewer Name: Edu Carballeira

Institution and Country: Gerontology Research Group

Department of Physiotherapy, Medicine & Biomedical Sciences. University of A Coruña

INIBIC- University Hospital Complex A Coruña (CHUAC)

Faculty of Health Sciences, Campus de Oza, 15006, A Coruña

Spain

Please state any competing interests or state ‘None declared’: None declared

Please leave your comments for the authors below

Comment - The present study protocol will assess the feasibility and efficacy of short duration (2-week), intensive (five-days per week) specially adapted exercise interventions within a delayed transfer of care hospital ward setting. I firmly believe that the aim of the study is a interesting topic in the area of health intervention in the geriatric population. Nevertheless, I have some concerns about the design of the study the should be addressed.

Response - Dear Dr. Carballeira, *Thank you for your review of our manuscript. Please find a response to all of your comments addressed below, and alterations highlighted within the manuscript as red text.*

Comment - General comments...

The main concern lies in the definition of the independent variable. The objective of the study is assess the feasibility and efficacy of two exercise protocol, one is an adapted machine-based resistance training intervention, and the other one a chair-based exercise intervention (Move It Or Lose It, MIOLI). Both protocols will last 35 min, however, duration is one of the multiple variables that defines training programme, and when a comparison between two exercise protocols is going to be made, the authors should define the differentiating variable (p.e. volume of muscles implicated) and

should explain how they are going to equate the rest of the variables (p.e. type of exercise, intensity, volume, density, recovery,...). There are many evidence that different combinations of the variables that defines training programme, the dose, influence on dependent variables, the response. That dose-response has to be well defined in order to know with some certainty what variable of the exercise protocol produced the outcomes observed in the dependent variables (physiological, functional, emotional, psychological / emotional, cognitive and social). For these reasons, I encourage the authors to apply two exercise protocols where the variable to be studied is well controlled, avoiding the confounding variables.

Response – *Just to clarify the Reviewer’s opening remarks as we are uncertain if this is a typing error, this study comprises of two separate independent variables to be examined within an independent measures research design as stated on Line 117 of the manuscript. Additionally, more accurately the primary aim of this study (lines 99 - 102) “is to assess the feasibility of a proposed future trial in this setting, which aims to assess the impact of specially adapted exercise interventions on the physiological, psychological, cognitive, social, and emotional health, and functional capacity of frail geriatric populations within a hospital ward setting”. The secondary aim of this feasibility study (lines 103 - 104) “is to assess the efficacy of the interventions on the primary dependent variables of the proposed future clinical trial within this setting.”*

As mentioned in the response to Reviewer 1, the length of the intervention sessions is primarily a projected estimation based on the exercises to be performed, rather than a chosen time period; it is simply the time it will take to complete the exercises, though dual concern was given to 1. The load and volume participants were asked to perform during each session, and 2. The overall length of the session. In this regard consideration was given to providing an adequate stimulus to participants to facilitate adaptation, while also not being too demanding as to cause undue fatigue throughout the intervention and to facilitate participant retention.

The authors feel that the differentiating internal components between the two exercise sessions are quite intricately outlined within figure 4 and 5 which consists of the entirety of the session protocols, in addition to lines 172 - 176, and lines 179 - 182 for the specially adapted resistance training intervention and the Move It Or Lose It intervention respectively (additive text in bold):

Lines 172 - 176:

*“This intervention will be comprised of an intensive (five days per week), short duration (2 week), 35 minutes per session, machine-based resistance training intervention. The exercises **performed** will specifically target the lower limbs **through a combination of strength and power training utilising a leg press and leg extension machine** (Figure 3). (Insert Figure 3)*

An outline of the protocol for each session can be found in Figure 4. (Insert Figure 4)”

Lines 179 - 182:

*“This intervention will comprise of an intensive (five days per week), short duration (2 week), **approximately** 35 minutes per session, chair-based exercise intervention. The exercises conducted will relate to strength, power, flexibility and aerobic capacity, predominantly targeting the lower limbs, but also incorporating the upper body and core. An outline of the protocol for each session can be found in Figure 5. (Insert Figure 5)*

Future to these intricate protocols we have also provided a detailed outline of the interventions as a whole in Table 2 on line 604, which is applicable to both interventions. This is also further supported on Lines 166 – 169 of the manuscript:

“All participants will undergo 10 sessions in total throughout the two-week intervention in either one of the two interventions. All session will be conducted individually under the guidance of a qualified trainer. A maximum of one session will be performed each day, and sessions will not be performed on any more than a maximum of three consecutive days throughout the duration of the study. (Table 2)”

As such the authors believe ample evidence has been provided regarding the two interventions in terms of the type of exercises being performed within each intervention and the predominant musculature involved as well as how confounding variables, not directly related to the internal content of the interventions, will be controlled with regard to volume, dose and recovery (though the authors are unsure what the Reviewer means when referring to “density?”), in addition to those variables listed within the manuscript on Lines 132 - 153, which are consistently implemented for each intervention. Variables external to the interventions / confounding variables not directly related to the internal components of the interventions are controlled in this regard; everything with the exception of the intrinsic content of the interventions which we wish to assess the feasibility and to a limited extent the efficacy of in order to inform the design of the future clinical trial.

We are attempting to assess the feasibility and to a limited extend the efficacy of these two separate established forms of interventions: one a relatively traditional (though adapted, particularly with regard to the equipment utilised) resistance training intervention, and the other an established community-based intervention for older adults. We are not attempting to compare two identical interventions, we are attempting to determine the feasibility of both, and to a degree the limited efficacy to inform the design of a future powered clinical trial. If the two intervention were perfectly similar in their internal components, we would not need to assess the feasibility of both as one would suffice, and how these two interventions differ internally is intricately detailed within the manuscript and will provide us feasibility data regarding both.

With regard to relative contributions of each component of an intervention to the impact on the dependent variables, which is what the authors believe the reviewer is articulating towards the end of the comment; as this is a non-powered feasibility study, we will not be able to determine the efficacy of the interventions within the study, only provide an estimate, and certainly not the efficacy of various components of the interventions. However, what we will be able to do is determine the feasibility of various components of the interventions and to a limited degree the overall efficacy of each individual intervention, that will then be assessed for efficacy within the powered future trail which is capable of determining this. However, it is important to note that we are not attempting to examine which modality or which component of the interventions produces the greatest relative effect in our secondary dependent variables, this is not within the scope of this study, nor would it be appropriate for this feasibility study. Our main concern as it relates to the interventions, are the feasibility of these two established separate forms of intervention as a whole, and the feasibility of each of the components which constitute them, while limited efficacy testing is a secondary concern regarding the intervention as a whole. As such the two independent variables are separate entities, we are not trying to make each the same as the other, we are trying to determine the feasibility of both within this setting.

Comment - Specific comments...

Abstract_Methods and analysis: details about independent variable should be provided.

Response - We have added this within methods and analysis section of the abstract as follows on lines 27 – 29 (additive text in bold):

*“This study will utilise a 2-week, interventional, independent measures research design in order to assess the feasibility and impact of two specially adapted exercise training interventions (**a specially adapted resistance training intervention, and Move It Or Lose It: an established community-***

based exercise intervention for older adults); aimed at improving the multi-dimensional health and functional capacity of frail geriatric hospital inpatients.”

Comment - Background_

page 4, line 5 Paraphrase the sentence, as is redundant.

Response – We have now reworded this sentence on Lines 68 – 69 as follows (altered text in bold):

“Exercise interventions have been proposed as potentially offering the best form of treatment for frail older adults (20); with exercise shown to be a significantly beneficial treatment for this population with regard to multiple components of health; and even shown to mediate the reversal of frailty in some cases (21-24)”

Comment - _page 6, line 22_ To state the differentiating variable between the two protocols.

Response – The authors trust this has been adequately addressed within the response to the reviewers initial comment pertaining to the internal components of each independent exercise intervention, illustrated in detail in figures 4 and 5.

Comment - Methods and analysis_ .

page 8, line 26 and figure 4 are the authors sure that repetition maximum is the best choice to evaluate the strength on geriatric hospitalized patients? Maybe other strength evaluations and monitoring tools could be employed, as maximum number of repetitions, velocity-based evaluations and velocity-based trainings, monitoring throughout perception of effort scales (i.e. OMNI-RPE monitoring). I encourage the authors to evaluate other options more suitable for this population, since the realization of the protocols of 1RM may lack reliability and may even be dangerous in this population.

Response – With regard to one repetition maximum assessments (1RM), this is based on an estimate from participant’s five repetition maximum (5RM) measurement, and thus far has been successfully implemented in all patients within the resistance training intervention without issue and rather seamlessly. It is worth noting that participants are performing 5 repetitions at 80% of their estimated 1RM within the intervention; typically the maximal load on a 5RM assessment will reach no more than 85% of participants 1RM. Additionally it is worth noting that all loads lifted during the course of the study are relative to the participants ability, and 5RM assessments are quite feasible. Though the Reviewer is correct that a direct 1RM assessment of course is generally not appropriate, advisable or necessary in any population other than young healthy individuals, hence an estimated 1RM based on 5RM was chosen. The Epley formula is then employed to provide an estimate of participants 1RM. In addition to providing a measure of maximal strength to act as a dependent variable, this is also used to calculate the load participants will lift within the specially adapted resistance training intervention as outlined in Figure 4 i.e. 50%, 60% and 80% of participants estimated 1RM for the warm up, power, and strength components of the intervention respectively. We have now added additional information regarding this to the manuscript on lines 259 - 261, as well as a supplementary file outlining the 5RM assessment protocol (Supplementary file 1):

“1RM will be determined based on estimation from participants five repetition maximum (5RM) utilising the Epley formula: $1RM = w(1 + r/30)$, where w = weight, and r = repetitions (38). The protocol employed during obtainment of participants 5RM has been adapted from (37) (Supplementary file 1).

Added manuscript references:

38. Epley, B. 1985, "Poundage chart", *Boyd Epley Workout*. Lincoln, NE: Body Enterprises, 2985, vol. 86, pp. p. 86.
37. Haff, G.G. & Triplett, N.T. 2015, *Essentials of strength training and conditioning 4th edition*, Human kinetics.

Comment - _page 10, line 50_ why these physiological variables are going to be analyzed? please explain your selection.

Response - *These physiological variables are being examined due to their association with the ageing process and previous research which has proposed and indicated that exercise is potentially capable of altering these variables. Please find a detailed description and justification of the inclusion of each of the physiological variables below:*

Cortisol: Cortisol is a stress hormone produced in the adrenal glands. Cortisol plays an essential role in the body's response to stress, but is generally immunosuppressing (Cupps, Fauci, 1982). However, if cortisol becomes either chronically elevated, or particularly low, this can lead to a plethora of negative health consequences, particularly if this elevation is not balanced by elevated levels of counteracting DHEA. Older adults are particularly susceptible to the relative increase in cortisol levels to DHEA, and the subsequent decline in immune system function (Segerstrom, Miller, 2004). This is predominantly due to the fact that DHEA gradually declines with age, a process known as adrenopause; while levels of cortisol remain relatively unaltered (Orentreich et al., 1992)

Dehydroepiandrosterone (DHEA): Dehydroepiandrosterone (DHEA) is an endogenous hormone produced predominantly by the adrenal cortex, and has been found to have immune enhancing properties (Orentreich et al., 1992). DHEA and its active sulphated form, DHEAS, has been proposed to be 'anti-ageing' (Chahal, Drake, 2007). However, DHEA declines progressively with age (reaching its peak between the ages of 20-30 years old), and by the age of approximately 70 can be as low as 10% of that in young age (Orentreich et al., 1992).

Serum cortisol: DHEAS ratio: As cortisol and DHEAS have opposing effects, it is often proposed that more important than the levels of either of these hormones alone, is the relative serum cortisol:DHEAS ratio. An increased cortisol:DHEA ratio results in the over expression of immunosuppressive glucocorticoids to immune enhancing DHEA; contributing to immunosenescence (Bauer, 2005, Butcher, Lord, 2004). While DHEAS levels fall continuously with age, cortisol levels remain relatively unaltered, resulting in a relative excess of cortisol over DHEAS (Orentreich et al., 1992). It has been postulated that exercise in older adults may be an effective method of enhancing DHEAS levels, similar to in younger adults (Phillips, Burns & Lord, 2007). DHEAS has been shown to increase after exercise in studies of post-menopausal women (Kemmler et al., 2003) and to be positively associated with physical activity performance in older men (O'Donnell et al., 2006), however other studies have shown no significant change in DHEAS in healthy older adults following exercise (Aldred et al., 2009). However, it has been consistently noted that further studies are required in older adults, particularly in older adults with various co-morbidities, in order to assess the response of DHEA and its sulphated form (DHEAS) to exercise within geriatric populations (Phillips, Burns & Lord, 2007, Aldred et al., 2009).

C-reactive protein (CRP): C-reactive protein is a plasma protein, the concentration of which increases both markedly and rapidly in the blood in response to inflammatory stimuli as part of the innate systemic immune response (Black, Kushner & Samols, 2004, Rosalki, 2001), and is therefore considered a generic marker of inflammation. Physical activity has been shown to significantly reduce circulating CRP levels within the blood (Goldhammer et al., 2005, Okita et al., 2004).

Inflammatory cytokines: Cytokines are small protein chemical messengers which are secreted by specific immune cells within the body. Cytokines travel via the circulatory system and ultimately have a subsequent effect on other cell elsewhere in the body (Zhang, An, 2007). Chronic systemic inflammation is associated with the development of a plethora of age and inactivity-related diseases and conditions such as atherosclerosis, diabetes and osteoarthritis (Argilés et al., 2003, Lechleitner et al., 2002, Smith, 2001). Inflammatory cytokines are a good measure of overall systemic inflammation status, and elevated blood concentrations of inflammatory cytokines and CRP are associated with increased age (Bruunsgaard, 2001, Rohde, Hennekens & Ridker, 1999). Physical activity and exercise have been shown to significantly reduced CRP and inflammatory cytokine levels within the blood (Goldhammer et al., 2005, Okita et al., 2004).

Interleukin 6 (IL-6): The anti-inflammatory properties of Interleukin 6 relate to the inhibition of the expression of TNF α , and indirectly blocking IL-1, both pro-inflammatory cytokines (Barton, 1997). IL-6 has been shown to be one of the most responsive cytokines to exercise, and has been proposed to have therapeutic potential in acute inflammation as a result (Barton, 1997, Ostrowski, K. et al., 1998, Ostrowski, Kenneth et al., 1999). Exercise induced production of IL-6 has been suggested to act as an anti-inflammatory cytokine, through blocking TNF α production (Starkie et al., 2003). IL-6 has been shown to be one of the most responsive cytokines to exercise, and has been proposed to have therapeutic potential in acute inflammation as a result, given its anti-inflammatory properties (Barton, 1997, Ostrowski, K. et al., 1998, Ostrowski, Kenneth et al., 1999). In relation to chronic inflammation, exercise training has been suggested to reduce the basal levels of IL-6 production and plasma IL-6 concentrations at rest, reducing the pro-inflammatory properties of IL-6 (Fischer, 2006).

Tumor Necrosis Factor alpha (TNF α): Tumor Necrosis Factor alpha is a pro-inflammatory cytokine involved in systemic inflammation. TNF α has been shown to upregulate mediators of inflammation such as nitric oxide and matrix metalloproteinases. Physically active individuals have a lower plasma concentration of TNF α compared to inactive individuals of the same age and gender (Reuben et al., 2003), and exercise has been shown to reduce basal levels of TNF α (Smart et al. 2011)

Interferon gamma (IFN γ): Interferon gamma (IFN γ) is a pro-inflammatory cytokine. Long term resistance training has been associated with reduced levels of IFN γ (a 45% reduction after 8-9 months of resistance training compared to baseline), in addition to reductions in IL-6 and TNF α (Córdova et al., 2011).

We have also now added the following rationale to the manuscript on lines 246 - 248:

"These specific physiological variables will be examined due to their association with the ageing process, and previous research which have both proposed and indicated that exercise is potentially capable of altering these variables (Heaney, Carroll & Phillips, 2013, Córdova et al., 2011, Smart et al., 2011, Fischer, 2006, Petersen, Pedersen, 2005, Goldhammer et al., 2005).

References:

Aldred, S., Rohalu, M., Edwards, K. & Burns, V. 2009, "Altered DHEA and DHEAS response to exercise in healthy older adults", J Aging Phys Act, vol. 17, no. 1, pp. 77-88.

Argilés, J.M., Moore-Carrasco, R., Fuster, G., Busquets, S. & López-Soriano, F.J. 2003, "Cancer cachexia: the molecular mechanisms", The international journal of biochemistry & cell biology, vol. 35, no. 4, pp. 405-409.

- Barton, B.E. 1997, "IL-6: Insights into Novel Biological Activities", *Clinical Immunology and Immunopathology*, vol. 85, no. 1, pp. 16-20.
- Bauer, M.E. 2005, "Stress, glucocorticoids and ageing of the immune system", *Stress*, vol. 8, no. 1, pp. 69-83.
- Black, S., Kushner, I. & Samols, D. 2004, "C-reactive protein", *Journal of Biological Chemistry*, vol. 279, no. 47, pp. 48487-48490.
- Bruunsgaard, H. 2001, "Effects of tumor necrosis factor-alpha and interleukin-6 in elderly populations.", *European cytokine network*, vol. 13, no. 4, pp. 389-391.
- Butcher, S.K. & Lord, J.M. 2004, "Stress responses and innate immunity: aging as a contributory factor", *Aging cell*, vol. 3, no. 4, pp. 151-160.
- Chahal, H.S. & Drake, W.M. 2007, "The endocrine system and ageing", *The Journal of pathology*, vol. 211, no. 2, pp. 173-180.
- Córdova, C., Lopes-e-Silva, J., Pires, A.S., Souza, V.C., Brito, C.J., Moraes, C.F., Sposito, A.C. & Nóbrega, O.T. 2011, "Long-Term Resistance Training Is Associated with Reduced Circulating Levels of IL-6, IFN-Gamma and TNF-Alpha in Elderly Women", *Neuroimmunomodulation*, vol. 18, no. 3, pp. 165-170.
- Cupps, T.R. & Fauci, A.S. 1982, "Corticosteroid-Mediated Immunoregulation in Man", *Immunological Reviews*, vol. 65, no. 1, pp. 133-155.
- Fischer, C.P. 2006, "Interleukin-6 in acute exercise and training: what is the biological relevance", *Exerc immunol rev*, vol. 12, no. 6-33, pp. 41.
- Goldhammer, E., Tanchilevitch, A., Maor, I., Beniamini, Y., Rosenschein, U. & Sagiv, M. 2005, "Exercise training modulates cytokines activity in coronary heart disease patients", *International journal of cardiology*, vol. 100, no. 1, pp. 93-99.
- Han, S.N. & Meydani, S.N. 2000, "Antioxidants, cytokines, and influenza infection in aged mice and elderly humans", *Journal of Infectious Diseases*, vol. 182, no. Supplement 1, pp. S80.
- Heaney, J.L., Carroll, D. & Phillips, A.C. 2013, "DHEA, DHEA-S and cortisol responses to acute exercise in older adults in relation to exercise training status and sex", *Age*, vol. 35, no. 2, pp. 395-405.
- Kemmler, W., Wildt, L., Engelke, K., Pintag, R., Pavel, M., Bracher, B., Weineck, J. & Kalender, W. 2003, "Acute hormonal responses of a high impact physical exercise session in early postmenopausal women", *European journal of applied physiology*, vol. 90, no. 1-2, pp. 199-209.
- Lechleitner, M., Herold, M., Dzien-Bischinger, C., Hoppichler, F. & Dzien, A. 2002, "Tumour necrosis factor-alpha plasma levels in elderly patients with Type 2 diabetes mellitus—observations over 2 years", *Diabetic Medicine*, vol. 19, no. 11, pp. 949-953.
- O'Donnell, A.B., Trivison, T.G., Harris, S.S., Tenover, J.L. & McKinlay, J.B. 2006, "Testosterone, dehydroepiandrosterone, and physical performance in older men: results from the Massachusetts Male Aging Study", *The Journal of Clinical Endocrinology & Metabolism*, vol. 91, no. 2, pp. 425-431.
- Okita, K., Nishijima, H., Murakami, T., Nagai, T., Morita, N., Yonezawa, K., Iizuka, K., Kawaguchi, H. & Kitabatake, A. 2004, "Can exercise training with weight loss lower serum C-reactive protein levels?", *Arteriosclerosis, Thrombosis, and Vascular Biology*, vol. 24, no. 10, pp. 1868-1873.

- Orentreich, N., Brind, J.L., Vogelmann, J.H., Andres, R. & Baldwin, H. 1992, "Long-term longitudinal measurements of plasma dehydroepiandrosterone sulfate in normal men.", *The Journal of Clinical Endocrinology & Metabolism*, vol. 75, no. 4, pp. 1002-1004.
- Ostrowski, K., Rohde, T., Zacho, M., Asp, S. & Pedersen, B.K. 1998, "Evidence that IL-6 is produced in skeletal muscle during intense long-term muscle activity", *J Physiol (Lond)*, vol. 508, pp. 949-953.
- Ostrowski, K., Rohde, T., Asp, S., Schjerling, P. & Pedersen, B.K. 1999, "Pro-and anti-inflammatory cytokine balance in strenuous exercise in humans", *The Journal of physiology*, vol. 515, no. 1, pp. 287-291.
- Petersen, A.M.W. & Pedersen, B.K. 2005, "The anti-inflammatory effect of exercise", *Journal of applied physiology*, vol. 98, no. 4, pp. 1154-1162.
- Phillips, A.C., Burns, V.E. & Lord, J.M. 2007, "Stress and exercise: getting the balance right for aging immunity", *Exercise and sport sciences reviews*, vol. 35, no. 1, pp. 35-39.
- Reuben, D.B., Judd-Hamilton, L., Harris, T.B. & Seeman, T.E. 2003, "The Associations Between Physical Activity and Inflammatory Markers in High-Functioning Older Persons: MacArthur Studies of Successful Aging", *Journal of the American Geriatrics Society*, vol. 51, no. 8, pp. 1125-1130.
- Rohde, L.E., Hennekens, C.H. & Ridker, P.M. 1999, "Survey of C-reactive protein and cardiovascular risk factors in apparently healthy men", *The American Journal of Cardiology*, vol. 84, no. 9, pp. 1018-1022.
- Rosalki, S.B. 2001, "C-reactive protein.", *International journal of clinical practice*, vol. 55, no. 4, pp. 269-270.
- Seegerstrom, S.C. & Miller, G.E. 2004, "Psychological stress and the human immune system: a meta-analytic study of 30 years of inquiry.", *Psychological bulletin*, vol. 130, no. 4, pp. 601.
- Smart, N.A., Larsen, A.I., Le Maitre, J.P. & Ferraz, A.S. 2011, "Effect of exercise training on interleukin-6, tumour necrosis factor alpha and functional capacity in heart failure", *Cardiology research and practice*, vol. 2011.
- Smith, J.K. 2001, "Exercise and atherogenesis", *Exercise and sport sciences reviews*, vol. 29, no. 2, pp. 49-53.
- Starkie, R., Ostrowski, S.R., Jauffred, S., Febbraio, M. & Pedersen, B.K. 2003, "Exercise and IL-6 infusion inhibit endotoxin-induced TNF- α production in humans", *The FASEB Journal*, vol. 17, no. 8, pp. 884-886.
- Zhang, J. & An, J. 2007, "Cytokines, Inflammation and Pain", *International anesthesiology clinics*, vol. 45, no. 2, pp. 27-37.

Comment - Statistical methods_

Secondary dependent variables_page 14, line 48_the authors must explain what analyzes would be carried out in case of outcomes do not fulfill the assumptions of normality.

Response - The authors have now added information to this effect on Lines 353 - 355:

“In the event data does not fulfill the assumptions of parametric statistics, a non-parametric version of the aforementioned statistical analysis will be employed, namely the Scheirer-Ray-Hare test, and Spearman’s rank-order correlations.”

VERSION 2 – REVIEW

REVIEWER	Freiberger Ellen Institute for Biomedicine of Aging Friedrich-Alexander-UNiversity ERlangen-Nürnberg/Germany
REVIEW RETURNED	04-Aug-2019

GENERAL COMMENTS	The authors are to congratulate for their improved manuscript. All points made by the reviewer have been addressed. The authors were right in their assumption that the reviewer did misread the main outcome (feasibility of the intervention). All explanation made by the authors are adequate and addressed the concerns of the reviewer. Only one minor concern remains. Not in congruence with the explanation of the authors, the reviewer would strongly suggest to delete the term “elderly” in the key words. In many other journals this terms is now even prohibited due to the negative impact and discrimination.
---

REVIEWER	Edu Carballeira Gerontology Research Group Department of Physiotherapy, Medicine & Biomedical Sciences.University of A Coruña INIBIC-University Hospital Complex A Coruña (CHUAC) Faculty of Health Sciences,Campus de Oza, 15006, A Coruña - Spain
REVIEW RETURNED	19-Aug-2019

GENERAL COMMENTS	I want to congratulate the authors for improving the paper with some suggested changes for both reviewers. However, I keep my concerns about the methodology of the the second objective of the paper "to asses the efficacy of two exercise interventions on secondary dependent variables". - 1RM as a reference parameter of strength: I don't understand why the authors think that 1RM load is the only load that can be employed to describe the strength of the participants. I recommend to review: Carpinelli. Med Sport. 2011. Strength characterization can be done throughout a rise in the load to a given number of repetitions or an increase of repetitions for a given load. - Estimation of 1RM throughout maximum number of reptitions has been validated employing mainly free weight exercises and with athletes. Furthermore, in these estimation protocols the failure has to be achieved to get a valid estimation. As I could check in the protocol described in Supplementary Material Figure 1, reaching failure seems not to be solicited to the participants, which on the other hand it is appreciated, since reaching the failure would be very stressful physiologically for the participants.
---

	- %1RM has limitations to prescribe exercise intensity as the number of repetitions that can be performed with a fixed %1RM is influenced by the amount of muscle mass used in the exercise (Shimano et al. J Strength Cond Res. 2006). I suggest the authors consider other alternatives as rating of perceived exertion (Helms et al. Front Physiol. 2018) to prescribe resistance training intensity. And even more taking into account that one of the exercise interventions has exercise wherel find it hard to imagine how 5RM was measured. - Two exercise interventions are going to be implemented, however I suggest the authors to check the typical strength training employed in older adults in the position statement from the National Strength and Conditioning Association in resistance training for older adults published recentley by Fragala et al. J Strength Cond Res. 2019. Since my point of view, the specially adpated resistance training intervention proposed in the present study does not represent the usual training programmed in these populations. - The authors indicate that comparison between exercise interventions is not the objective of the study. A 2x2 ANOVA is going to be applied checking for interaction and post-hoc analysis, therefore, if the parameters that define both interventions are not controlled it would be diffciult to extract conclusions of the results. When we speak about parameters defining resistance training we are referring to muscle mass implicated, load intensity and effort intensity (Steele J. Br J Sports Med. 2014), volume (work or series x reps x distance), recovery, density (work:pause ratio), even distribution of pauses (Iglesias-Soler et al. Clin Physiol Funct Imaging. 2014) all of them affect the physiological variables. Therefore, even being a secondary objective, the two exercise interventions should have some of these parameters equated, p.e. a dose-response relationship in physical activity and the muscle mass implicated have demonstrated effects on C-reactive protein and TNF-a (Fragala et al. J Strength Cond Res. 2019).
--	--

VERSION 2 – AUTHOR RESPONSE

Editorial comments:

The reviewer(s) have recommended revisions to your manuscript. Therefore, I invite you to respond to the reviewer(s)' comments and revise your manuscript. Please note that we normally allow a maximum of two manuscript revisions. As such, we urge you to make all the necessary revisions at this stage in an effort to convince the reviewers that your work is suitable for publication in BMJ Open.

Dear Editor,

Thank you again for your review of our manuscript and permitting us to submit this revised version. We are pleased that we have adequately addressed all of the Editor's initial comments. Please find our response to the remaining two Reviewer's comments below.

Reviewer: 1

Reviewer Name: Freiburger Ellen

Institution and Country: Institute for Biomedicine of Aging Friedrich-Alexander-UNiversity Erlangen-Nürnberg/Germany

Please state any competing interests or state 'None declared': None declared

Comment: The authors are to congratulate for their improved manuscript. All points made by the reviewer have been addressed. The authors were right in their assumption that the reviewer did misread the main outcome (feasibility of the intervention). All explanation made by the authors are adequate and addressed the concerns of the reviewer.

Only one minor concern remains. Not in congruence with the explanation of the authors, the reviewer would strongly suggest to delete the term "elderly" in the key words. In many other journals this term is now even prohibited due to the negative impact and discrimination.

Response: *Dear Dr. Frieberger,*

Thank you again for your review of our manuscript. We acknowledge and appreciate the minor concern expressed by the Reviewer regarding the inclusion of the adjective "elderly" within the study's key words. However, while the authors respectfully maintain their initial position, it has come to the authors attention that BMJ Open do not publish a key words section within their manuscripts; as such we have removed this section from the manuscript in its entirety.

Reviewer: 2

Reviewer Name: Edu Carballeira

Institution and Country:

Gerontology Research Group

Department of Physiotherapy, Medicine & Biomedical Sciences. University of A Coruña

INIBIC-University Hospital Complex A Coruña (CHUAC)

Faculty of Health Sciences, Campus de Oza, 15006, A Coruña - Spain

Please state any competing interests or state 'None declared': None declared

Please leave your comments for the authors below

Comment: I want to congratulate the authors for improving the paper with some suggested changes for both reviewers. However, I keep my concerns about the methodology of the the second objective of the paper "to asses the efficacy of two exercise interventions on secondary dependent variables".

Response: Dear Dr. Carballeira,

Thank you again for your review of our manuscript. Please find a response to each of your comments, primarily regarding the remaining concerns about the methodology of the secondary aim of this feasibility study, which we have now altered slightly on line 103 – 104 of the manuscript to specifically reflect the limited efficacy testing within this component of the paper. These lines now read as follows (added text in bold):

Line 103 – 104:

*“The secondary aim of this feasibility study is to assess the **potential** efficacy of the interventions on the primary dependent variables of the proposed future clinical trial within this setting”.*

Comment: - 1RM as a reference parameter of strength:

I don't understand why the authors think that 1RM load is the only load that can be employed to describe the strength of the participants. I recommend to review: Carpinelli. Med Sport. 2011. Strength characterization can be done throughout a rise in the load to a given number of repetitions or an increase of repetitions for a given load.

Response: *The authors agree with the Reviewer that a strength assessment can be conducted in many ways; including through an increase in load at a given number of repetitions, or an increase of repetitions at a given load. Our rationale for the choice of 1RM as the reference parameter for maximal strength within this present feasibility study is that it has been consistently shown to be a safe, valid, and reliable measure of maximal strength in older adults (Barbalho et al., 2018, Rydwik et al., 2007, Knutzen, BRILLA & CAINE, 1999, Shaw, McCully & Posner, 1995); widely utilised as a reference value for the prescription of strength and power training within this population (Martínez-Velilla et al., 2019, Daly et al., 2013, Lovell, Cuneo & Gass, 2010, Venturelli et al., 2010, Granacher, Gruber & Gollhofer, 2009, Kalapotharakos et al., 2007, DiFrancisco-Donoghue, Werner & Douris, 2007, Bottaro et al., 2007, Fatouros, Ioannis G. et al., 2006, Henwood, Taaffe, 2006, Beneka et al., 2005, De Vos et al., 2005, Fatouros, I. G. et al., 2005, Seynnes et al., 2004, Vincent et al., 2002, Sullivan et al., 2001, Hortobágyi et al., 2001, Hunter et al., 2001, Fiatarone et al., 1994, Charette et al., 1991, Fiatarone et al., 1990), even recently in hospitalised older adults (Martínez-Velilla et al., 2019), and is considered the gold standard in this regard (Fragala et al., 2019, Haff, Triplett, 2015, Miller, 2012).*

Moreover, the recently published position statement from the National Strength and Conditioning Association regarding resistance training for older adults, also supports the use of 1RM measurements as the maximal strength reference parameter for prescribing resistance training intensity in older adults: “Resistance training intensity should be based on a percentage of estimated 1RM” (Fragala et al., 2019 page 2030). This evidence-based position statement also advocates the utilisation of relative loads for “frail” older adults consistent with those employed within this present feasibility study (Strength - 80% 1RM, Power - 60% 1RM). However, the authors acknowledge that these guidelines, 1) were published prior to the commencement of the study and submission of this protocol manuscript; as such having no direct influence on its design; 2) predominantly pertain to community dwelling older adults and older adults in nursing homes rather than geriatric hospital inpatients, (although for the latter, no such guidelines exist) and; 3) largely reference “frail” in a descriptive, rather than in an operationally defined manner.

The authors thank the Reviewer for the recommendation of the narrative review by Carpinelli 2011 – “Assessment of one repetition maximum (1RM) and 1RM prediction equations: are they really necessary?”. However, the authors note that this narrative review presents no critique or argument against the safety, validity, or reliability of the utilisation of 1RM measurements, or the, validity, or reliability of 1RM estimating equations, to provide a reference value for maximal strength. Rather the

review poses whether there are alternatives which can potentially be employed i.e. are 1RM assessments or 1RM prediction equations absolutely essential to defining maximal strength. As such it is not a critique of the safety, validity or reliability of using 1RM, but rather the essential necessity, and if alternatives can potentially be employed, which we agree, they potentially can be.

The authors are aware of several studies which have chosen to implement for example the Borg scale, as the reference parameter for exercise intensity, in pre-sarcopenic community dwelling older adults (Vikberg et al., 2019), and healthy community dwelling older women (De Vreede et al., 2005). However, there are a number of other issues with alternatives, such as rate of perceived exertion scales, specifically regarding the characteristics of the participants within this present feasibility study i.e. frail geriatric delayed transfer of care patients, often presenting with dementia or neurological disorders. In this regard objective measures (such as estimated 1RM) are more appropriate than subjective measures (rate of perceived exertion); in which the feasibility and validity of the latter could be easily called into question in this regard within this setting and patient cohort.

Comment: - Estimation of 1RM throughout maximum number of repetitions has been validated employing mainly free weight exercises and with athletes. Furthermore, in these estimation protocols the failure has to be achieved to get a valid estimation. As I could check in the protocol described in Supplementary Material Figure 1, reaching failure seems not to be solicited to the participants, which on the other hand it is appreciated, since reaching the failure would be very stressful physiologically for the participants.

Response: We believe there may be a slight misunderstanding here. The authors are not employing “an estimation of 1RM throughout maximum number of repetitions”, rather, or more specially, the authors are employing an estimation of 1RM load, derived from participants maximum 5RM load, utilising the Epley formula (Epley, 1985), to prescribe exercise intensities, which are consistent with that of the recommended values for strength (80% 1RM), and power (60% 1RM) training (Haff, Triplett, 2015), employed within the specially adapted resistance training intervention. These values are also consistent with the recently published position statement from the National Strength and Conditioning Association regarding resistance training for older adults; including resistance training for “frail” older adults (Fragala et al., 2019).

With reference to the Reviewer’s point about the estimation of maximal strength through 1RM measurements, we agree that while being validated mainly using free weights in younger populations, it has also been consistently validated using machine based exercises in older adults; including each of the specific machine based exercises utilised within this present feasibility study (Barbalho et al., 2018, Phillips et al., 2004, Knutzen, BRILLA & CAINE, 1999). Additionally, machine based 1RM assessments have also been consistently shown to be reliable in the maximal strength assessment of older adults; including again specifically for the two machine based exercises performed within this present feasibility study (Martínez-Velilla et al., 2019, Granacher, Gruber & Gollhofer, 2009, Bottaro et al., 2007, DiFrancisco-Donoghue, Werner & Douris, 2007, Kalapotharakos et al., 2007, Fatouros, Ioannis G. et al., 2006, Henwood, Taaffe, 2006, Beneka et al., 2005, De Vos et al., 2005, Fatouros, I. G. et al., 2005, Vincent et al., 2002, Hortobágyi et al., 2001, Hunter et al., 2001, Rooks et al., 1997, Charette et al., 1991).

We would also politely disagree with the Reviewer that these estimation protocols require the achievement of muscular failure in all case to obtain a valid estimation in older adults. Rather there are two scenarios where a valid estimation can be obtained, 1) the achievement of momentary muscular failure 2) the achievement of volitional failure - a mutually agreed consensus that more weight cannot be added and a lack of will of the participant to continue in this regard. Both are valid and appropriate grounds for concluding a maximal strength test in older adults (Phillips et al., 2004).

As is evident in Supplementary file 1, both volitional or momentary muscular failure are solicited to the participants between Step 6a – Step 6c:

“Step 6a - If the participant is successful, and both the researcher and the participant feel this is the participants 5RM, use this value as the participants 5RM.

Step 6b - If the participant is successful and feels as if they can add more weight, provide 2 minutes rest and add 5 -10 % of the existing load.

Step 6c - In the event a participant is unable to perform 5 repetitions at the existing weight in Step 5, remove 5% of existing load and repeat step 5”

Step 6a in the above regard does not necessarily need to be stated, as it is implied in Step 6b, where the grounds for continuing the 1RM assessment are 1) if the participant is successful, and 2) if they feel they can add more weight. In the event the latter is not present, then this is consistent with the added Step 6a which was added to highlight that volitional failure is afforded to these frail geriatric patients before momentary muscular failure if at all possible.

Comment: - %1RM has limitations to prescribe exercise intensity as the number of repetitions that can be performed with a fixed %1RM is influenced by the amount of muscle mass used in the exercise (Shimano et al. J Strength Cond Res. 2006). I suggest the authors consider other alternatives as rating of perceived exertion (Helms et al. Front Physiol. 2018) to prescribe resistance training intensity. And even more taking into account that one of the exercise interventions has exercise where I find it hard to imagine how 5RM was measured.

Response: *With regard to the first aspect of the Reviewer’s comment, the stated limitation of using %1RM as a reference value to prescribe exercise intensity is not applicable in this instance as the 1RM is directly measured for these exact exercises performed within the intervention (i.e. leg press and leg extension), and the relative loads of 60% 1RM (power training) and 80% 1RM (strength training) subsequently applied to these two exact exercises based on their respective estimated 1RM. Hence it is the same exercise being performed during the assessment of 1RM, and within the specially adapted resistance training intervention i.e. leg press and leg extension; therefore, the same musculature is utilised within the 1RM assessment, as within the resistance training intervention.*

With regard to the second aspect of the Reviewers comment:

“I suggest the authors consider other alternatives as rating of perceived exertion (Helms et al. Front Physiol. 2018) to prescribe resistance training intensity.”

The rate of perceived exertions is a predominantly subjective measurement, while assessment of 1RM is a predominantly objective measurement and the gold standard for maximal strength assessment in older adults, consistently shown to be a safe, valid, and reliable measurement in this regard. On these grounds alone 1RM is the superior measure relative to the rate of perceived exertion. Additionally, this is a feasibility study in frail geriatric delayed transfer of care patients, often presenting with dementia and neurological disorders. In this regard objective measures are more appropriate where possible than subjective measures, the feasibility and validity of the latter of which may be called into question in this regard. The referenced study cited by the Reviewer consists of a sample of young healthy resistance trained participants (18 – 35 years), which is a cohort requiring much different considerations compared with this present feasibility study. Although, the authors are aware of the utilisation of the rate of perceived exertion in other studies in healthy community dwelling older adults (Vikberg et al., 2019, De Vreede et al., 2005), these also represent a cohort with different considerations to those within this present feasibility study.

With regard to the third aspect of the Reviewer’s comment:

“And even more taking into account that one of the exercise interventions has exercise where I find it hard to imagine how 5RM was measured.”

The authors presume the Reviewer is referencing the Move It Or Lose It (MIOLI) intervention in this instance, if so, we would like to reiterate that - 1RM maximum measurements are conducted only on the leg press and leg extension machines, and then used to inform the load lifted on both the leg press and leg extension machine respectively, solely within the specially adapted resistance training intervention. As such it is directly applicable because it is the exact same exercise. 1RM measurements are not applied to the Move It Or Lose It intervention, because they are not applicable in this regard. Rather a screening at the start of the MIOLI intervention is conducted to determine whether participants should perform exercises standing, standing with chair support, or seated. With regard to any exercises using resistance training bands, participants have a choice of three resistances which are assessed prior to intervention commencement, as described on lines 269 – 275 of the manuscript, which we have now altered to avoid any ambiguity in this regard (added text in bold):

Lines 269 – 275:

“A balance screening will also be conducted prior to the commencement of the MIOLI intervention to determine whether participants should perform exercises standing, standing with chair support, or seated. In addition, the resistance bands which participants in the MIOLI intervention will use will similarly be assessed during this period; with three options corresponding to light, moderate and high resistances, prescribed to participants based on their initial ability and preference during their performance of the exercises in which the resistance bands will be utilised”.

Comment: - Two exercise interventions are going to be implemented, however I suggest the authors to check the typical strength training employed in older adults in the position statement from the National Strength and Conditioning Association in resistance training for older adults published recently by Fragala et al. J Strength Cond Res. 2019. Since my point of view, the specially adapted resistance training intervention proposed in the present study does not represent the usual training programmed in these populations.

Response: *The authors apologise, as having checked Figure 4, which outlines the protocol of the specially adapted exercise intervention, we have noticed a typing error within the figure; namely the main intervention was initially reported as:*

“Main Intervention (Strength and power training) – Leg extension: 2 sets of 8 reps at 60% 1RM, Leg extension 2 sets of 5 reps at 80% 1RM, Leg press 1 set of 8 reps at 60% 1RM, Leg press 1 set of 5 reps at 80% 1RM”.

Rather, Figure 4 should read as follows:

“Main intervention (Strength and power training) – Leg Press 1 sets of 8 reps at 60% 1RM, Leg Press 2 sets of 5 reps at 80% 1RM, Leg extension 1 sets of 8 reps at 60% 1RM, Leg extension 2 sets of 5 reps at 80% 1RM”.

We have now corrected this within Figure 4 and resubmitted along with this response letter and amended protocol manuscript.

The specially adapted resistance training intervention for frail geriatric delayed transfer of care patients employed within this present feasibility study is largely consistent with these guidelines in that participants perform 3 sets of 2 multi-joint lower-body exercises (leg press and leg extension) at 80% of their estimated one repetition maximum (1RM) within the predominantly strength training

component of the resistance training intervention; and 60% 1RM at a higher velocity within the predominantly power training component of the resistance training intervention. Additionally, 1 set of warm up is performed at 50% 1RM on the leg extension and leg press machines.

However, the authors do note that no guidelines are provided within this position statement for geriatric hospital inpatients, mainly because there is a profound lack of research in this area, which the authors of this present study are attempting to address with this mixed methods feasibility study. The authors additionally note that although loads up to 80% 1RM for strength training, and 60% for power training are recommended within these guidelines, that the evidence used to derive these recommendations are more often than not based on studies where participants have been descriptively referred to as frail, rather than operationally defined as frail. As such, particularly given the former, even with the high degree of consistency between the specially adapted resistance intervention employed within this feasibility study, and these guidelines, the premise that there exists a usual training programme for frail geriatric hospital inpatients (not to mention delayed transfer of care patients) is unfounded, and again highlights the importance of research such as this present feasibility study.

While noting that the above referenced position statement from the NSCA regarding resistance training in older adults 1) was published subsequent to the commencement of this present feasibility study, and submission of this protocol manuscript; 2) predominantly pertains to community dwelling older adults and older adults in nursing homes, rather than geriatric hospital inpatients; 3) largely references “frail” in a descriptive, rather than in an operationally defined manner, - to further strengthen this component of the manuscript, the authors have now made reference to the high degree of consistency between the specially adapted exercise intervention utilised within this present feasibility study and the National Strength and Conditioning Association position statement for resistance training in “frail” older adults on Lines 171 - 181 of the manuscript as follows (added text in bold):

Lines 171 – 181:

“Exercise Intervention 1: Specially-adapted resistance training intervention

*This intervention will be comprised of an intensive (five days per week), short duration (2 week), approximately 35 minutes per session, machine-based resistance training intervention. The exercises performed will specifically target the lower limbs through a combination of **multi-joint** strength and power training utilising a leg press and leg extension machine (Figure 3). (Insert Figure 3)*

The maximal strength reference value (% 1RM), duration (35 minutes), type of exercise (multi-joint), loadings (60% 1RM (Power), 80% 1RM (Strength)), and volume (3 sets, 5 – 8 repetitions) are largely consistent with the position statement from the National Strength and Conditioning Association (NSCA) regarding resistance training in older adults (Fragala et al. 2019)

An outline of the protocol for each session can be found in Figure 4. (Insert Figure 4)”

We believe the fact that the specially adapted resistance training intervention employed within this feasibility study is consistent with these guidelines illustrates the high quality and intricate detail of all elements of the protocol manuscript, even the minutia of niche components of this feasibility study. However, to further clarify in this regard, the authors chose 80% 1RM as the intensity of the strength training component of the intervention as it is within the range of 70 – 85% proposed as the optimal range for strength adaption for older adults (Haff, Triplett, 2015), and consistent with previous research in older adults, which have shown that greater intensities with regard to 1RM, in the range of 80%+ 1RM, are consistently shown to result in greater improvements in strength and mental health measures in older adults when compared to lower 1RM intensities (Steib, Schoene & Pfeifer, 2010, Singh et al., 2005). Participants in this present study also commenced the intervention with loads of

50% for warm up, 60% for power training and 80% for strength training, no periodisation, was employed, and similarity 1RM was not reassessed during the duration of the study, due to the 2 week maximal length of the intervention. Additionally, exercise sessions were employed 5 times a week in these patients to attempt to provide the greatest benefit within the relatively short period of time. This is consistent with previous research in hospitalised older adults which has shown that training even twice per day, for short durations, can be well tolerated in acute hospitalised older adults (Martínez-Velilla et al., 2019). As such we believe the specially adapted resistance training intervention employed within this present study is largely consistent with that of general resistance training interventions employed in older adults, however, note that a relatively large range exists in this regard.

Comment: - The authors indicate that comparison between exercise interventions is not the objective of the study. A 2x2 ANOVA is going to be applied checking for interaction and post-hoc analysis, therefore, if the parameters that define both interventions are not controlled it would be difficult to extract conclusions of the results. When we speak about parameters defining resistance training we are referring to muscle mass implicated, load intensity and effort intensity (Steele J. Br J Sports Med. 2014), volume (work or series x reps x distance), recovery, density (work:pause ratio), even distribution of pauses (Iglesias-Soler et al. Clin Physiol Funct Imaging. 2014) all of them affect the physiological variables. Therefore, even being a secondary objective, the two exercise interventions should have some of these parameters equated, p.e. a dose-response relationship in physical activity and the muscle mass implicated have demonstrated effects on C-reactive protein and TNF-a (Fragala et al. J Strength Cond Res. 2019).

Response:

We would like to clarify again that as this is a feasibility study, the design of which has been largely been informed by Bowen et al., 2009 – “How we design feasibility studies”, the primary aim of this study, as stated on lines 99 – 103 (and further elaborated upon on lines 108 – 110) of the manuscript, is to:

Line 99 – 103:

“assess the feasibility of a proposed future trial in this setting, which aims to assess the impact of specially adapted exercise interventions on the physiological, psychological, cognitive, social, and emotional health, and functional capacity of frail geriatric populations within a hospital ward setting; recognising health as a multi-factorial concept incorporating inter-related dimensions.”

Lines 108 – 110:

“Assessment of the feasibility of the study as it relates to the eight-primary areas of focus for feasibility studies (acceptability, demand, implementation, practicality, adaptation, integration, expansion and limited-efficacy testing)”

The authors would like to reemphasise that a feasibility study is quite different from a randomised controlled trial (or indeed a pilot study) in terms of its aims and objectives, and as such its overall design as a means to achieve these aims and objectives. Respectfully, the authors think it is possible the Reviewer may have misinterpreted this aspect of the feasibility study as being akin to a more typical comparative study in sports science between two training modalities. However, the authors would like to clarify again that this is not the case; rather we are seeking to establish the feasibility and to a limited degree the efficacy of the two established forms of interventions within this setting. With regard to the latter we do not wish to compare two different modalities of training, but rather a limited assessment of the impact of the two established programmes, each slightly adapted for the setting (while still maintaining their intrinsic nature), and in so a limited comparison of these two

established programmes through our limited efficacy testing. We hope that further to our initial response, this adequately clarifies this aspect of the manuscript for the Reviewer.

We once again thank the Editor and all the Reviewers for their time in the review of this manuscript and hope that all the above comments have been adequately addressed in a clear and logical manner, and that the manuscript is now suitable for publication.

Yours Sincerely,

Mr. Paul Doody, Dr. Carolyn Greig, Professor Janet Lord, Professor Anna Whittaker

References:

- Barbalho, M., Gentil, P., Raiol, R., Del Vecchio, F., Ramirez-Campillo, R. & Coswig, V. 2018, "High 1RM Tests Reproducibility and Validity are not Dependent on Training Experience, Muscle Group Tested or Strength Level in Older Women", *Sports*, vol. 6, no. 4.
- Beneka, A., Malliou, P., Fatouros, I., Jamurtas, A., Gioftsidou, A., Godolias, G. & Taxildaris, K. 2005, "Resistance training effects on muscular strength of elderly are related to intensity and gender", *Journal of science and medicine in sport*, vol. 8, no. 3, pp. 274-283.
- Bottaro, M., Machado, S.N., Nogueira, W., Scales, R. & Veloso, J. 2007, "Effect of high versus low-velocity resistance training on muscular fitness and functional performance in older men", *European journal of applied physiology*, vol. 99, no. 3, pp. 257-264.
- Bowen, D.J., Kreuter, M., Spring, B., Cofta-Woerpel, L., Linnan, L., Weiner, D., Bakken, S., Kaplan, C.P., Squiers, L. & Fabrizio, C. 2009, "How we design feasibility studies", *American Journal of Preventive Medicine*, vol. 36, no. 5, pp. 452-457.
- Charette, S., McEvoy, L., Pyka, G., Snow-Harter, C., Guido, D., Wiswell, R.A. & Marcus, R. 1991, "Muscle hypertrophy response to resistance training in older women", *Journal of applied physiology*, vol. 70, no. 5, pp. 1912-1916.
- Daly, M., Vidt, M.E., Eggebeen, J.D., Simpson, W.G., Miller, M.E., Marsh, A.P. & Saul, K.R. 2013, "Upper extremity muscle volumes and functional strength after resistance training in older adults", *Journal of Aging and Physical Activity*, vol. 21, no. 2, pp. 186-207.
- De Vos, N.J., Singh, N.A., Ross, D.A., Stavrinou, T.M., Orr, R. & Fiatarone Singh, M.A. 2005, "Optimal load for increasing muscle power during explosive resistance training in older adults", *The Journals of Gerontology Series A: Biological Sciences and Medical Sciences*, vol. 60, no. 5, pp. 638-647.
- De Vreede, P.L., Samson, M.M., Van Meeteren, N.L., Duursma, S.A. & Verhaar, H.J. 2005, "Functional-task exercise versus resistance strength exercise to improve daily function in older women: a randomized, controlled trial", *Journal of the American Geriatrics Society*, vol. 53, no. 1, pp. 2-10.

- DiFrancisco-Donoghue, J., Werner, W. & Douris, P.C. 2007, "Comparison of once-weekly and twice-weekly strength training in older adults", *British journal of sports medicine*, vol. 41, no. 1, pp. 19-22.
- Epley, B. 1985, "Poundage chart", *Boyd Epley Workout*. Lincoln, NE: Body Enterprises, 2985, vol. 86, pp. p. 86.
- Fatouros, I.G., Kambas, A., Katrabasas, I., Nikolaidis, K., Chatzinikolaou, A., Leontsini, D. & Taxildaris, K. 2005, "Strength training and detraining effects on muscular strength, anaerobic power, and mobility of inactive older men are intensity dependent", *British journal of sports medicine*, vol. 39, no. 10, pp. 776-780.
- Fatouros, I.G., Kambas, A., Katrabasas, I., Leontsini, D., Chatzinikolaou, A., Jamurtas, A.Z., Douroudos, I., Aggelousis, N. & Taxildaris, K. 2006, "Resistance training and detraining effects on flexibility performance in the elderly are intensity-dependent", *The Journal of Strength & Conditioning Research*, vol. 20, no. 3, pp. 634-642.
- Fiatarone, M.A., Marks, E.C., Ryan, N.D., Meredith, C.N., Lipsitz, L.A. & Evans, W.J. 1990, "High-Intensity Strength Training in Nonagenarians: Effects on Skeletal Muscle", *ResearchGate*, vol. 263, no. 22, pp. 3029-34.
- Fiatarone, M.A., O'Neill, E.F., Ryan, N.D., Clements, K.M., Solares, G.R., Nelson, M.E., Roberts, S.B., Kehayias, J.J., Lipsitz, L.A. & Evans, W.J. 1994, "Exercise training and nutritional supplementation for physical frailty in very elderly people", *New England Journal of Medicine*, vol. 330, no. 25, pp. 1769-1775.
- Fragala, M.S., Cadore, E.L., Dorgo, S., Izquierdo, M., Kraemer, W.J., Peterson, M.D. & Ryan, E.D. 2019, "Resistance Training for Older Adults: Position Statement From the National Strength and Conditioning Association", *The Journal of Strength & Conditioning Research*, vol. 33, no. 8.
- Granacher, U., Gruber, M. & Gollhofer, A. 2009, "Resistance training and neuromuscular performance in seniors", *International Journal of Sports Medicine*, vol. 30, no. 09, pp. 652-657.
- Haff, G.G. & Triplett, N.T. 2015, *Essentials of strength training and conditioning 4th edition*, Human kinetics.
- Henwood, T.R. & Taaffe, D.R. 2006, "Short-term resistance training and the older adult: the effect of varied programmes for the enhancement of muscle strength and functional performance", *Clinical physiology and functional imaging*, vol. 26, no. 5, pp. 305-313.
- Hortobágyi, T., Tunnel, D., Moody, J., Beam, S. & DeVita, P. 2001, "Low-or high-intensity strength training partially restores impaired quadriceps force accuracy and steadiness in aged adults", *The Journals of Gerontology Series A: Biological Sciences and Medical Sciences*, vol. 56, no. 1, pp. B38-B47.
- Hunter, G.R., Wetzstein, C.J., McLafferty, J.C., Zuckerman, P.A., Landers, K.A. & Bamman, M.M. 2001, "High-resistance versus variable-resistance training in older adults.", *Medicine and science in sports and exercise*, vol. 33, no. 10, pp. 1759-1764.
- Kalappotharakos, V.I., Smilios, I., Parlavatzas, A. & Tokmakidis, S.P. 2007, "The effect of moderate resistance strength training and detraining on muscle strength and power in older men", *Journal of Geriatric Physical Therapy*, vol. 30, no. 3, pp. 109-113.
- Knutzen, K.M., BRILLA, L.R. & CAINE, D. 1999, "Validity of 1RM prediction equations for older adults", *The Journal of Strength & Conditioning Research*, vol. 13, no. 3, pp. 242-246.

- Lovell, D.I., Cuneo, R. & Gass, G.C. 2010, "The effect of strength training and short-term detraining on maximum force and the rate of force development of older men", *European journal of applied physiology*, vol. 109, no. 3, pp. 429-435.
- Martínez-Velilla, N., Casas-Herrero, A., Zambom-Ferraresi, F., de Asteasu, Mikel L Sáez, Lucia, A., Galbete, A., García-Baztán, A., Alonso-Renedo, J., González-Glaría, B. & Gonzalo-Lázaro, M. 2019, "Effect of exercise intervention on functional decline in very elderly patients during acute hospitalization: a randomized clinical trial", *JAMA internal medicine*, vol. 179, no. 1, pp. 28-36.
- Miller, T.A. 2012, *NSCA's Guide to Tests and Assessments*, Human Kinetics.
- Phillips, W.T., Batterham, A.M., Valenzuela, J.E. & Burkett, L.N. 2004, "Reliability of maximal strength testing in older adults", *Archives of Physical Medicine and Rehabilitation*, vol. 85, no. 2, pp. 329-334.
- Rooks, D.S., Kiel, D.P., Parsons, C. & Hayes, W.C. 1997, "Self-paced resistance training and walking exercise in community-dwelling older adults: effects on neuromotor performance", *The Journals of Gerontology Series A: Biological Sciences and Medical Sciences*, vol. 52, no. 3, pp. M161-M168.
- Rydwik, E., Karlsson, C., Frändin, K. & Akner, G. 2007, "Muscle strength testing with one repetition maximum in the arm/shoulder for people aged 75 -test-retest reliability", *Clinical rehabilitation*, vol. 21, no. 3, pp. 258-265.
- Seynnes, O., Fiatarone Singh, M.A., Hue, O., Pras, P., Legros, P. & Bernard, P.L. 2004, "Physiological and functional responses to low-moderate versus high-intensity progressive resistance training in frail elders", *The Journals of Gerontology Series A: Biological Sciences and Medical Sciences*, vol. 59, no. 5, pp. M503-M509.
- Shaw, C.E., McCully, K.K. & Posner, J.D. 1995, "Injuries during the one repetition maximum assessment in the elderly.", *Journal of cardiopulmonary rehabilitation*, vol. 15, no. 4, pp. 283-287.
- Singh, N.A., Stavrinou, T.M., Scarbek, Y., Galambos, G., Liber, C., Fiatarone Singh, M.A. & Morley, J.E. 2005, "A randomized controlled trial of high versus low intensity weight training versus general practitioner care for clinical depression in older adults", *The Journals of Gerontology: Series A*, vol. 60, no. 6, pp. 768-776.
- Steib, S., Schoene, D. & Pfeifer, K. 2010, "Dose-response relationship of resistance training in older adults: a meta-analysis", *Medicine & Science in Sports & Exercise*, vol. 42, no. 5, pp. 902-914.
- Sullivan, D., Wall, P., Tim PhD, P.T., Bariola, J., Bopp, M. & Frost, Y. 2001, , *Progressive Resistance Muscle Strength Training of Hospitalized Frail Elderly* [Homepage of From the Geriatric Research Education and Clinical Center, Central Arkansas Veterans Healthcare System (DHS), and the Donald W. Reynolds Department of Geriatrics (DHS, JRB, MMB, YMF) and the Department of Physical Medicine and Rehabilitation (PTW), University of Arkansas for Medical Sciences, Little Rock, Arkansas], [Online]. Available: <http://ovidsp.ovid.com/ovidweb.cgi?T=JS&PAGE=reference&D=ovfte&NEWS=N&AN=00002060-200107000-00007780>].
- Venturelli, M., Lanza, M., Muti, E. & Schena, F. 2010, "Positive effects of physical training in activity of daily living-dependent older adults", *Experimental aging research*, vol. 36, no. 2, pp. 190-205.
- Vikberg, S., Sörlén, N., Brandén, L., Johansson, J., Nordström, A., Hult, A. & Nordström, P. 2019, "Effects of resistance training on functional strength and muscle mass in 70-year-old individuals with pre-sarcopenia: a randomized controlled trial", *Journal of the American Medical Directors Association*, vol. 20, no. 1, pp. 28-34.

Vincent, K.R., Braith, R.W., Feldman, R.A., Magyari, P.M., Cutler, R.B., Persin, S.A., Lennon, S.L., Md, A.H.G. & Lowenthal, D.T. 2002, "Resistance exercise and physical performance in adults aged 60 to 83", *Journal of the American Geriatrics Society*, vol. 50, no. 6, pp. 1100-1107.

VERSION 3 – REVIEW

REVIEWER	Freiberger Ellen FAU Erlangen-Nürnberg institute for Biomedicine of Aging Germany
REVIEW RETURNED	01-Oct-2019
GENERAL COMMENTS	All comments have been addressed and the reviewer is satisfied.